# Basonuclin-2 promotes fracture repair through NuRD-dependent chromatin remodeling in periosteal stem cells

Zhong Zhang [ID] [1,2,4], Lingli Zhang[3,4], Bo Jiang[1,2,4], Shuqin Chen[1], Wenhui Xing[1], Peilong Wang[2], Lixiang Lou[2], Chunxiao Tang[1], Xuye Hu[2], Jinlong Suo [ID] [2], Bo O Zhou [ID] [2], Weiguo Zou [ID] [1,2 ✉] & Lijun Wang [ID] [1 ✉]

## Abstract

**Bone fracture healing remains a significant challenge in orthopedics, as injury-responsive skeletal stem cell (SSC) populations and the regulatory mechanisms governing SSC activation during nonunion fracture repair remain poorly delineated. This study identifies zinc finger transcription factor basonuclin-2 (BNC2) as a skeletal fracture repair control factor in periosteal stem cells. BNC2 marks quiescent periosteal cells during homeostasis and is significantly upregulated upon injury in mice, driving endochondral ossification post-fracture via clonal expansion. Moreover, knockout of *Bnc2* in *Prx1-cre*[+] cells (not *Ocn-cre*[+] osteoblasts or *LepR-creER*[+] BMSCs) resulted in impaired fracture healing, suppressing SSC proliferation. Mechanistically, ATAC-seq revealed that BNC2 deletion reduced chromatin accessibility at promoter regions of proliferation genes, hindering transcriptional activation. Additionally, BNC2 regulates histone H3 acetylation by interacting with the NuRD complex. Pharmacologically inhibition of HDAC1/2 activity partially ameliorated the fracture repair defects observed in *Prx1-cre; Bnc2[f/f]* mice. Collectively, we identified BNC2[+] cells as a rapidly expanding periosteal cell population inducing endochondral ossification niches during repair, providing potential therapeutic strategies for nonunion fractures.**

**Keywords** Periosteal Skeletal Stem Cells; Fracture Healing; *Bnc2*; NuRD Complex
**Subject Categories** Chromatin, Transcription & Genomics; Musculoskeletal System

## Introduction

Bone fractures are the most common large organ traumatic injury in humans. Fracture healing is a postnatal regenerative process that recapitulates multiple events in embryonic skeletal development (Bais et al, 2009). Fracture repair usually restores the injured skeletal site to its pre-injury cellular composition, structure, and biomechanical function. However, there are still 5–10% of fractures that do not heal properly. The causes of nonunion are multifactorial, including the original trauma, the means of treatment, and the patient's intrinsic factors, suggesting that both physical and biological factors can interfere with normal fracture healing (Wildemann et al, 2021). Stem cell function is the most important part of the patient's intrinsic factors. Stem cell therapy is an important means to promote nonunion healing. Through specific cytokine treatment, endogenous stem cells could be recruited to the site of injury and further promote cell proliferation, migration, adhesion and differentiation (Tepper et al, 2005). SSCs in different parts of the bone have different molecular characteristics (Feng et al, 2022). For instance, *Prx1* labels SSCs in both the bone marrow and periosteum (Liu et al, 2022), *Ctsk* specifically marks periosteal SSCs (Debnath et al, 2018), and *LepR* serves as a marker for SSCs localized within the bone marrow (Jeffery et al, 2022). Although multiple skeletal stem cell populations have been identified, it remains unclear whether there are skeletal stem cells that respond to injury repair without affecting homeostasis.

Epigenetic modification plays an important role in bone remodeling and bone repair. Knockout of the DNA methylase *Dnmt3b* in chondrocytes leads to delayed fracture repair and endochondral ossification (Wang et al, 2018). Deletion of H3K36 methylase NSD1 or H3K4 demethylase LSD1 in mesenchymal stem cells can lead to impaired fracture repair (Shao et al, 2021; Sun et al, 2020). During the differentiation process of osteoblasts, the acetylation levels of histones H3 and H4 in the promoter region of osteoblast-related marker genes increased significantly, and the knockdown of HDAC1 or treated with HDAC inhibitor trichostatin A (TSA) could enhance the differentiation of osteoblasts (Cho et al, 2014; Lee et al, 2006). In addition, TSA could promote human dental pulp stem cell proliferation and odontoblast differentiation (Jin et al, 2013). Inhibitors of BET family proteins can be used as drugs to target epigenetic modifications to promote bone repair (Chen et al, 2019). However, the regulatory mechanism of HDAC on fracture repair needs to be further explored.

[1]Hainan Institute of Regenerative Orthopedics and Sports Medicine, Key Laboratory of Emergency and Trauma of Ministry of Education, Hainan Academy of Medical Sciences, Hainan Medical University, Hainan 571199, China. [2]Key Laboratory of RNA Innovation, Science and Engineering, CAS Center for Excellence in Molecular Cell Science, Shanghai Institute of Biochemistry and Cell Biology, University of Chinese Academy of Sciences, Chinese Academy of Sciences, Shanghai 200031, China. [3]School of Athletic Performance, Shanghai University of Sport, Shanghai 200438, China. [4]These authors contributed equally: Zhong Zhang, Lingli Zhang, Bo Jiang. ✉E-mail: zouwg94@sibcb.ac.cn; wanglijun2014@sibcb.ac.cn

Basonuclin-2 (BNC2) is a zinc finger protein that is predominantly expressed in germ cells, which is also widely expressed in skeletal systems, such as ear cartilage, nasal cartilage, intervertebral disc nucleus pulposus, and perichondrium of long bones. *Bnc2* knockout mice have a body size reduction phenotype, and knockout of *Bnc2* resulted in a reduced craniofacial mesenchymal cell proliferation (Vanhoutteghem et al, 2009). Marie Bobowski-Gerard et al, found that *BNC2* expression was significantly promoted during the hepatic fibrosis process, and BNC2 could play a role in promoting fibrosis by activating TGFβ and Hippo/YAP1 pathways (Bobowski-Gerard et al, 2022). However, whether *Bnc2* can affect bone repair and its regulatory mechanisms remains unknown.

Here, we found that the expression of *Bnc2* was significantly upregulated in SSCs at the early stage after bone fracture by transcriptome sequencing. BNC2 could label periosteal cells, and these cells were fully involved in endochondral osteogenesis during fracture healing through lineage tracing with *Bnc2-creER; Rosa26-LSL-tdTomato* reporter mice, which also indicated the specific function of *Bnc2* in periosteal SSCs. To further explore the regulatory effect of *Bnc2* on fracture healing, we constructed *Bnc2* conditional knockout mice lines in different cell populations. We found that *Prx1-cre; Bnc2^{f/f}* mice had significant impairments in fracture repair, while this phenotype was not present in *Ocn-cre; Bnc2^{f/f}* and *LepR-creER; Bnc2^{f/f}* mice, demonstrating that BNC2 played roles in periosteal SSCs. Mechanistically, BNC2 was indispensable for SSC proliferation upon injury. BNC2 regulated the level of H3ac of callus SSCs by interacting with the NuRD complex, thereby affecting chromatin accessibility to regulate gene transcription activation and cell proliferation.

# Results

## *Bnc2* expression is upregulated in the periosteal SSCs after fracture

In order to explore the key transcription factors that regulate SSC activation in the early stage of fracture healing, we obtained SSCs from uninjured periosteum and 3 dpf callus (3 days post-fracture) by flow cytometry for bulk RNAseq (Fig. 1A). The flow cytometry results showed that the proportion of SSCs at the callus was significantly upregulated, indicating that the periosteal SSCs underwent significant proliferation (Fig. 1A,B). By performing KEGG enrichment analysis of genes upregulated in SSCs of periosteal callus, we found that SSCs in callus were significantly increased the expression of genes associated with cell cycle and DNA replication (Appendix Fig. S1A). There were five transcription factors in the significantly upregulated genes (Fig. 1C; Appendix Fig. S1B), among which *E2f3* promoted cell proliferation by regulating the transition from G1 to S phase (Leone et al, 1998), and *Bnc2* was reported to regulate normal meiosis in spermatogonial stem cells (Vanhoutteghem et al, 2014). Single-cell sequencing of *Prx1-cre; Rosa26-LSL-tdTomato* positive periosteal cells showed that *Bnc2* was specifically expressed in skeletal precursor cells (Appendix Fig. S2A–C) (Wang, Ren et al, 2025; Data ref: Wang, Ren et al, 2025). The expression of *Bnc2* in periosteal SSCs was significantly higher than that in bone marrow SSCs (Appendix Fig. S2D). QRT-PCR confirmed that *Bnc2* expression was significantly

upregulated in the 3 dpf callus, compared with uninjured periosteum (Fig. 1D). To further investigate the expression pattern of *Bnc2* during fracture repair, we constructed a *Bnc2-P2A-EGFP* mouse model. *Bnc2-EGFP* positive cells existed in the periosteum and significantly increased upon injury (Fig. 1E,F). Flow cytometry analysis of 3 dpf callus cells from *Bnc2-P2A-EGFP* mice showed that the proportion of SSCs in *Bnc2-EGFP* positive cells was significantly higher than that in *Bnc2-EGFP* negative cells (Fig. 1G,H). However, the absolute number of *Bnc2^+* SSCs in the periosteum was less than that of *Bnc2^-* SSCs (Fig. 1I), since there are far more BNC2^- than BNC2^+ in the Lin^-6C3^-Thy1^- gate. The above results demonstrate that *Bnc2* is specifically expressed in periosteal SSCs in the early stage of fracture repair and is likely to regulate the proliferation of SSCs.

## BNC2 could label periosteal cells

As *Bnc2* was highly expressed in the SSCs of the periosteum (Appendix Fig. S2A–D), we sought to explore whether the specificity effect of *Bnc2* on periosteal SSCs is due to its expression pattern. We constructed *Bnc2-creER; Rosa26-LSL-tdTomato; Col1-GFP* (*Col2.3-GFP*) mice and treated the newborn mice with tamoxifen (Fig. 2A). Short-term labeling showed that *Bnc2-creER* specifically label the periosteal cells, including the periosteum near the growth plate and in the diaphyseal segment, with a few labeling in the metaphyseal cancellous bone region and no labeling at all in the bone marrow cavity of the diaphyseal segment (Fig. 2B). For adult *Bnc2-creER; Rosa26-LSL-tdTomato; Col1-GFP* mice, short-term labeling also showed that *Bnc2-creER* could specifically label periosteal cells (Fig. 2C,D). Long-term tracing showed similar results, except that the periosteum was more adequately labeled. However, periosteal *Bnc2-creER* positive cells did not transform into *Col1-GFP* positive osteoblasts in long-term tracing (Fig. 2E,F). By flow cytometry analysis of *Bnc2-creER; Rosa26-LSL-tdTomato* mice, we also confirmed that BNC2 mainly labeled periosteal cells, less labeled for metaphysis region, and hardly labeled for bone marrow in diaphysis region (Fig. 2G–I). These data suggest that BNC2 could label periosteal cells, and BNC2-positive periosteal cells are in a relatively resting state and do not contribute to osteogenesis in homeostasis.

## BNC2-positive periosteal cells contribute to endochondral ossification during fracture healing

Since the expression of *Bnc2* was significantly higher in SSCs from callus than in periosteal SSCs (Fig. 1C,D), we sought to explore whether BNC2-positive cells could be involved in fracture healing. We performed fracture surgery in *Bnc2-creER; Rosa26-LSL-tdTomato; Col1-GFP* mice (Fig. 3A). *Bnc2-creER* positive cells could fully label the proliferated periosteal cells on 3 dpf (Fig. 3B), and contribute to callus chondrocytes on 7 dpf (Fig. 3C). Besides, a small number of osteoblasts formed by intramembranous ossification at the distal part of callus were *Bnc2-creER* positive (Fig. 3C). Furthermore, *Bnc2-creER* positive cells could fully differentiate into osteoblasts formed by endochondral ossification in callus (Fig. 3D), but only made a very small contribution to the osteoblasts of callus in bone marrow on 14 dpf (Fig. 3D). After 56 days post-fracture, *Bnc2-creER* positive cells could form neo-bone at the healing site of fracture (Fig. 3E). At the distal end of the fracture, *Bnc2-creER*

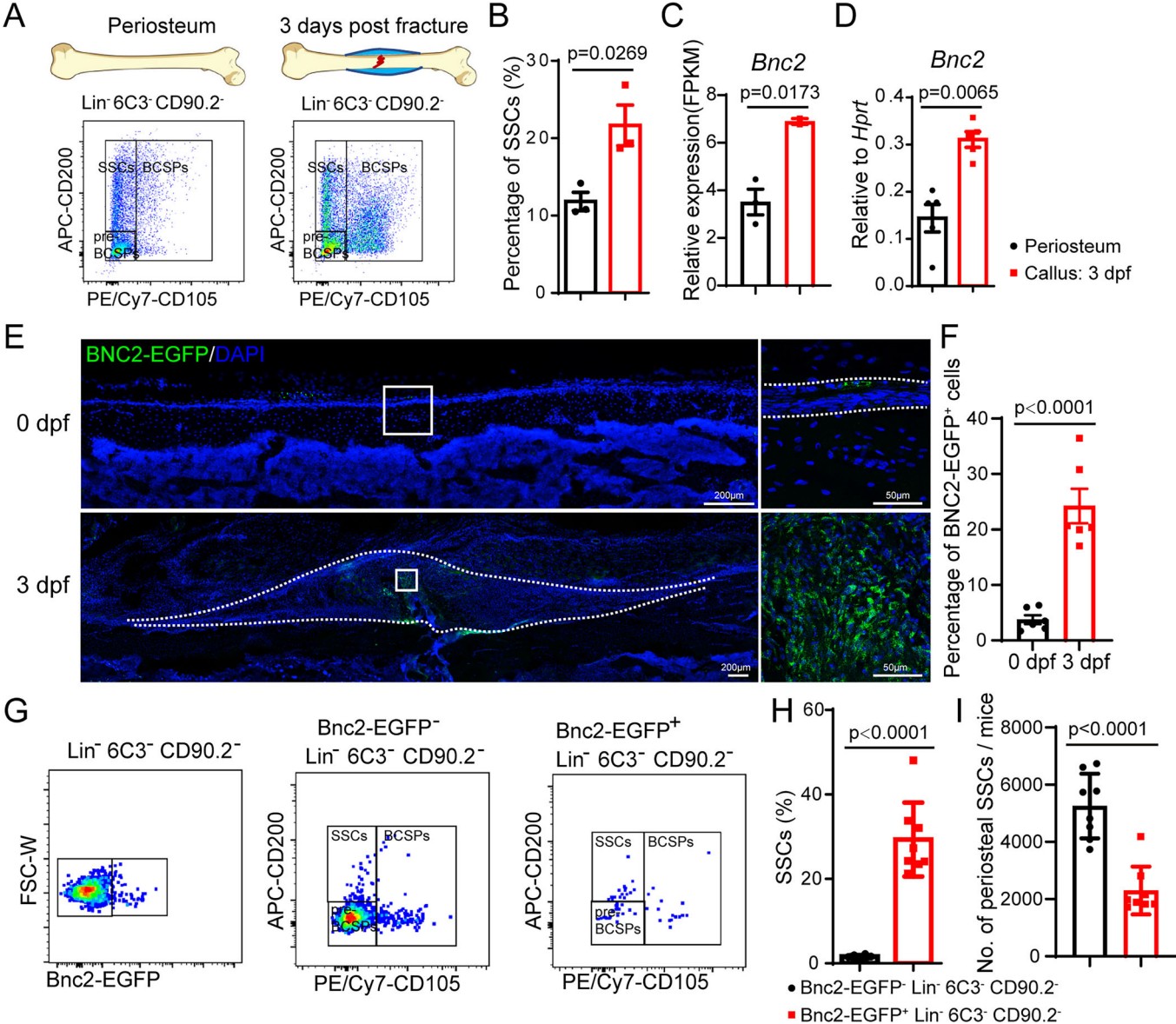

**Figure 1. Expression of *Bnc2* is upregulated in the periosteal SSCs after fracture.**

(A, B) SSCs flow cytometry results (A) and quantitative statistical results (B) of callus cells on 3 dpf and undamaged periosteum. Data were presented as the means ± SEM. n = 3. Unpaired t-test. BCSP, bone cartilage stromal progenitor. (C) Expression of *Bnc2* in RNAseq of 3 dpf callus and periosteal SSCs. Data were presented as the means ± SEM. n = 3, 2. Unpaired t-test. (D) Undamaged periosteum and 3 dpf callus SSCs were sorted, and qRT-PCR was used to detect the expression of *Bnc2*. Data were presented as the means ± SEM. n = 5. Unpaired t-test. (E, F) The expression of *BNC2-EGFP* on 0 dpf and 3 dpf, and the quantitative statistics of the proportion of *BNC2-EGFP* positive cells. The area within the dotted line is the callus. Data were presented as the means ± SEM. n = 6. Unpaired t-test. (G) Flow cytometry clustering of SSCs in *BNC2-EGFP* negative and positive periosteal cells. (H, I) Quantitative statistics of percentage and number of SSCs in *BNC2-EGFP* negative and positive periosteal cells. Data were presented as the means ± SEM. n = 8. Unpaired t-test. Source data are available online for this figure.

positive cells still only labeled the periosteal cells and did not contribute to bone formation at uninjured site (Fig. 3E). Then we tried to investigate whether *Bnc2-creER* positive periosteal cells participate in intramembrane osteogenesis during injury repair. We performed a bone drilling injury model with *Bnc2-creER; Rosa26-LSL-tdTomato; Col1-GFP* mice (Fig. 3F). BM SSCs, rather than periosteal SSCs, were reported to contribute to new bone formation in bone drilling injury model (Jeffery et al, 2022). Consistently, *Bnc2-creER* positive cells could still adequately label periosteal cells

near the injured site, but had little contribution to osteoblasts formed at the injured site (Fig. 3G).

*LepR* has been shown to label BM SSCs and mainly contribute to the formation of callus within the bone marrow(Jeffery et al, 2022; Shu et al, 2021). Our studies in *LepR-creER; Rosa26-LSL-tdTomato; Col1-GFP* mice further confirmed that *LepR-creER* predominantly labels trabecular bone and BM SSCs (Fig. 4A,B). We observed that *Bnc2-creER⁺* cells made very little contribution to the callus in the bone marrow (Fig. 3D). To further confirm the specificity of *Bnc2*

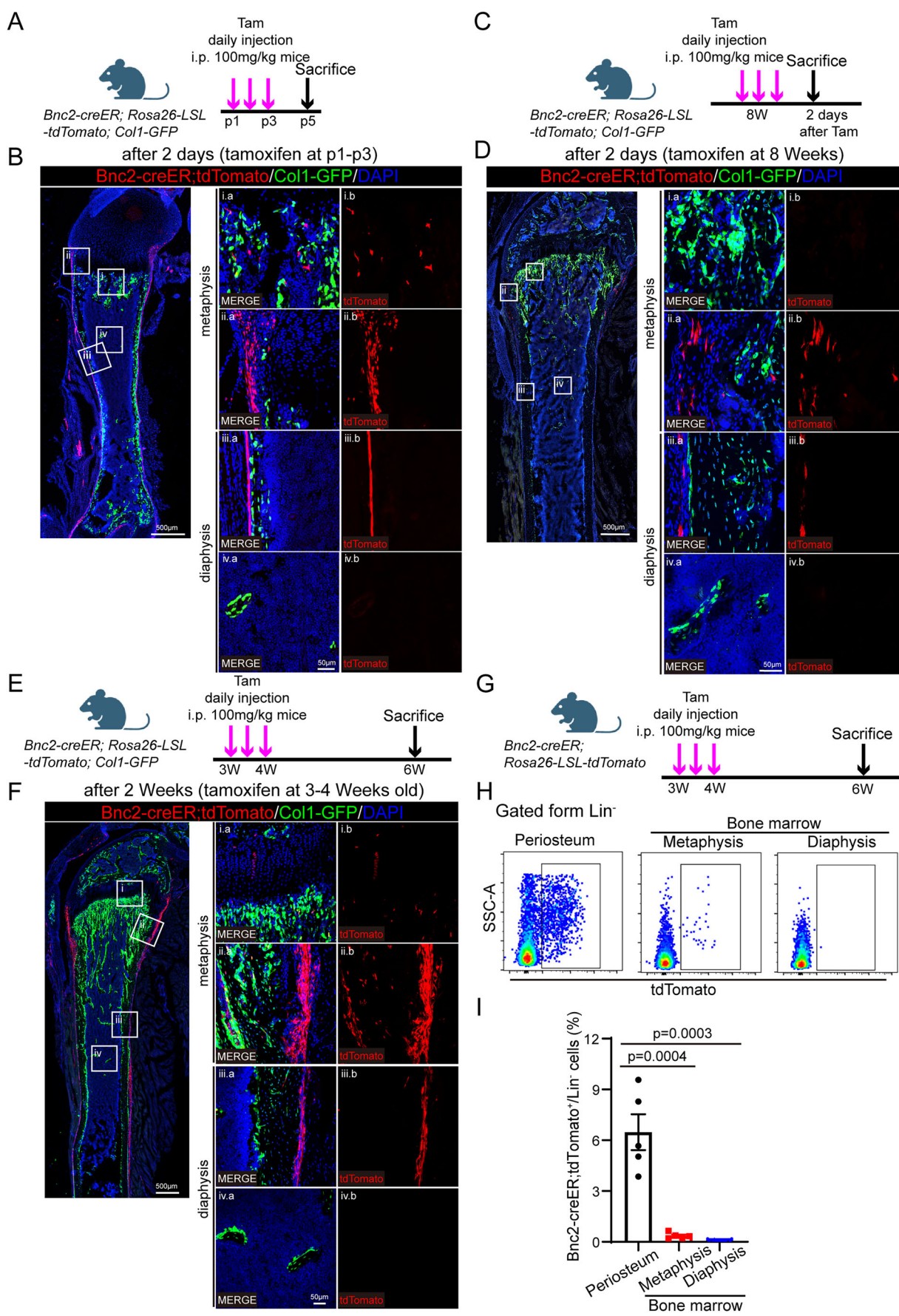

Figure 2. *Bnc2-creER* specifically labels periosteum in homeostasis.

(A, B) Lineage tracing of neonatal *Bnc2-creER; Rosa26-LSL-tdTomato; Col1-GFP* mice at 2 days after tamoxifen induction. (C, D) Lineage tracing of 8-week-old *Bnc2-creER; Rosa26-LSL-tdTomato; Col1-GFP* mice at 2 days after tamoxifen induction. (E and F) Lineage tracing of 4-week-old *Bnc2-creER; Rosa26-LSL-tdTomato; Col1-GFP* mice at 2 weeks after tamoxifen induction. (G) Schematic diagram of tamoxifen induction strategy in *Bnc2-creER; Rosa26-LSL-tdTomato* mice for flow cytometry. (H, I) The proportion and quantitative statistics of *Bnc2-creER; Rosa26-LSL-tdTomato* labels in periosteum, metaphyseal and diaphyseal bone marrow of long bone. Data were presented as the means ± SEM. $n = 5$. Unpaired *t*-test. Source data are available online for this figure.

for periosteal cell labeling, *LepR-dreER; Rosa26-RSR-GFP* reporter mice was mated with *Bnc2-creER; Rosa26-LSL-tdTomato* mice to obtain dual-reporter mice. Fracture modeling was performed on *Bnc2-creER; Rosa26-LSL-tdTomato; LepR-dreER; Rosa26-RSR-GFP* mice (Fig. 4C). Utilizing dual-reporter mouse models, we demonstrated that under uninjured conditions, *Bnc2-creER* exclusively labels periosteal cells, whereas *LepR-dreER* labels trabecular bone and BM SSCs (Fig. 4D). During fracture healing, the specificity of *Bnc2-creER; Rosa26-LSL-tdTomato* positive cells' contribution to the external callus was demonstrated, and *LepR-dreER; Rosa26-RSR-GFP* positive cells mainly formed the internal callus in bone marrow, and a small amount of *LepR-dreER; Rosa26-RSR-GFP* positive cells was found in the external callus. However, there was no colocalization of *Bnc2-creER; Rosa26-LSL-tdTomato* and *LepR-dreER; Rosa26-RSR-GFP* positive cells in the external callus (Fig. 4E,F).

Our findings demonstrate that BNC2 could label a subpopulation of periosteal SSCs that exhibit robust engagement in endochondral ossification during fracture repair. These BNC2+ cells orchestrate the regeneration of functional periosteal tissue and cortical bone architecture critical for structural restoration, while showing negligible contribution to bone marrow-derived internal callus formation during the healing process.

## Loss of *Bnc2* in periosteal SSCs shows significant impairment in fracture healing

To explore whether *Bnc2* is necessary for periosteal SSCs function in bone repair, we delete *Bnc2* in SSCs/osteoblasts/osteocytes using *Prx1-cre; Bnc2^f/f^* mice, and in BM SSCs using *LepR-creER; Bnc2^f/f^* mice (Fig. 5A,B). X-Ray analysis showed that the callus volume was significantly smaller in *Prx1-cre; Bnc2^f/f^* mice but not in *LepR-creER; Bnc2^f/f^* mice (Fig. 5C,D). The fracture nonunion and reduced newly formed bone volume was observed in *Prx1-cre; Bnc2^f/f^* mice but not in *LepR-creER; Bnc2^f/f^* mice, as assessed by micro-CT scanning and quantification (Fig. 5E–G). Consistently, Safranine O/Fast green (SOFG) and immunostaining of COL2 demonstrated that loss of *Bnc2* in *Prx1-cre+* but not in *LepR-creER+* cells impaired endochondral osteogenesis during fracture healing (Fig. 5H,I). In order to explore whether *Bnc2* deficiency in *Prx1-cre+* cells affects intramembrane osteogenesis during repair, *Prx1-cre; Bnc2^f/f^* mice were subjected to a bone drilling injury model, as this injury was repaired by intramembranous ossification (Colnot, 2009). SOFG staining and OPN (Osteopontin) immunofluorescence staining showed that *Prx1-cre; Bnc2^f/f^* mice could normally form trabecular bone at the site of injury 14 days after surgery (Appendix Fig. S3A,B). Micro-CT scanning results also showed that there was no significant difference between *Prx1-cre; Bnc2^f/f^* and control mice during drilling damage repair (Appendix Fig. S3C,D).

To dissect the cell type-specific functions of BNC2 in fracture healing, we employed complementary genetic strategies: To eliminate the influence of the extensive expression of *Prx1-Cre* during development (SSCs/osteoblasts/osteocytes), we used the *Prx1-CreER* mouse model to knock out *Bnc2* at adulthood, thereby achieving the knockout of *Bnc2* in periosteal cells in the cortical bone region. X-ray analysis showed a decreased callus volume of *Prx1-creER; Bnc2^f/f^* mice during the fracture healing process (Appendix Fig. S4A–C). The micro-CT results showed that there was an obvious nonunion of bone fracture in *Prx1-creER; Bnc2^f/f^* mice (Appendix Fig. S4D,E). Besides, we used the *Ocn-cre; Bnc2^f/f^* mice to selectively ablate *Bnc2* in mature osteo-lineage cells. Quantitative analysis revealed comparable callus volumes between *Ocn-cre; Bnc2^f/f^* mice and controls during fracture healing (Appendix Fig. S5A,B), with micro-CT at 4 weeks post-fracture confirming complete bony union in both groups (Appendix Fig. S5C,D). Histological assessment via SOFG staining and COL2 immunostaining demonstrated normal endochondral ossification dynamics in *Ocn-cre; Bnc2^f/f^* mice callus (Appendix Fig. S5E,F). Moreover, tamoxifen-inducible *Ocn-creER; Rosa26-LSL-tdTomato* lineage tracing exhibited strict exclusion from periosteal stem cell pools and chondrogenic lineages throughout healing phases (Appendix Fig. S6A–C).

These data establish a functional dichotomy: *Prx1-cre; Bnc2^f/f^* and *Prx1-creER; Bnc2^f/f^* mice display arrested healing, whereas both *LepR-creER; Bnc2^f/f^* and *Ocn-cre; Bnc2^f/f^* mice models maintain normal repair kinetics. This genetic epistasis analysis unequivocally identifies periosteal SSC-derived BNC2 as an indispensable regulator of fracture healing.

## *Bnc2* deficiency inhibits SSC activation in the early stage of fracture healing

The periosteal thickening response in *Prx1-cre; Bnc2^f/f^* mice was significantly impaired on 3 dpf (Appendix Fig. S7A), thus we sought to explore whether the impairment of fracture healing in *Prx1-cre; Bnc2^f/f^* mice begin with early obstruction of periosteal cell proliferation. For this, the proportion of SSCs in *Prx1-cre; Bnc2^f/f^* mice was analyzed by flow cytometry (Fig. 6A). The proportion of SSCs significantly reduced in the 3 dpf callus but not in the uninjured periosteum of *Prx1-cre; Bnc2^f/f^* mice, compared with littermate control (Fig. 6B,C). To further confirm the effect of *Bnc2* on SSC activation, we labeled *Prx1-cre; Bnc2^f/f^* mice with EdU injection (Fig. 6D). The proportion of EdU+ precursor cells significantly reduced in the 3 dpf callus but not in the uninjured periosteum of *Prx1-cre; Bnc2^f/f^* mice, compared with littermate control (Fig. 6E,F). Additionally, EdU staining results also showed that *Prx1-cre; Bnc2^f/f^* mice developed disorders at the stage of periosteum thickening, and the proportion of EdU-positive

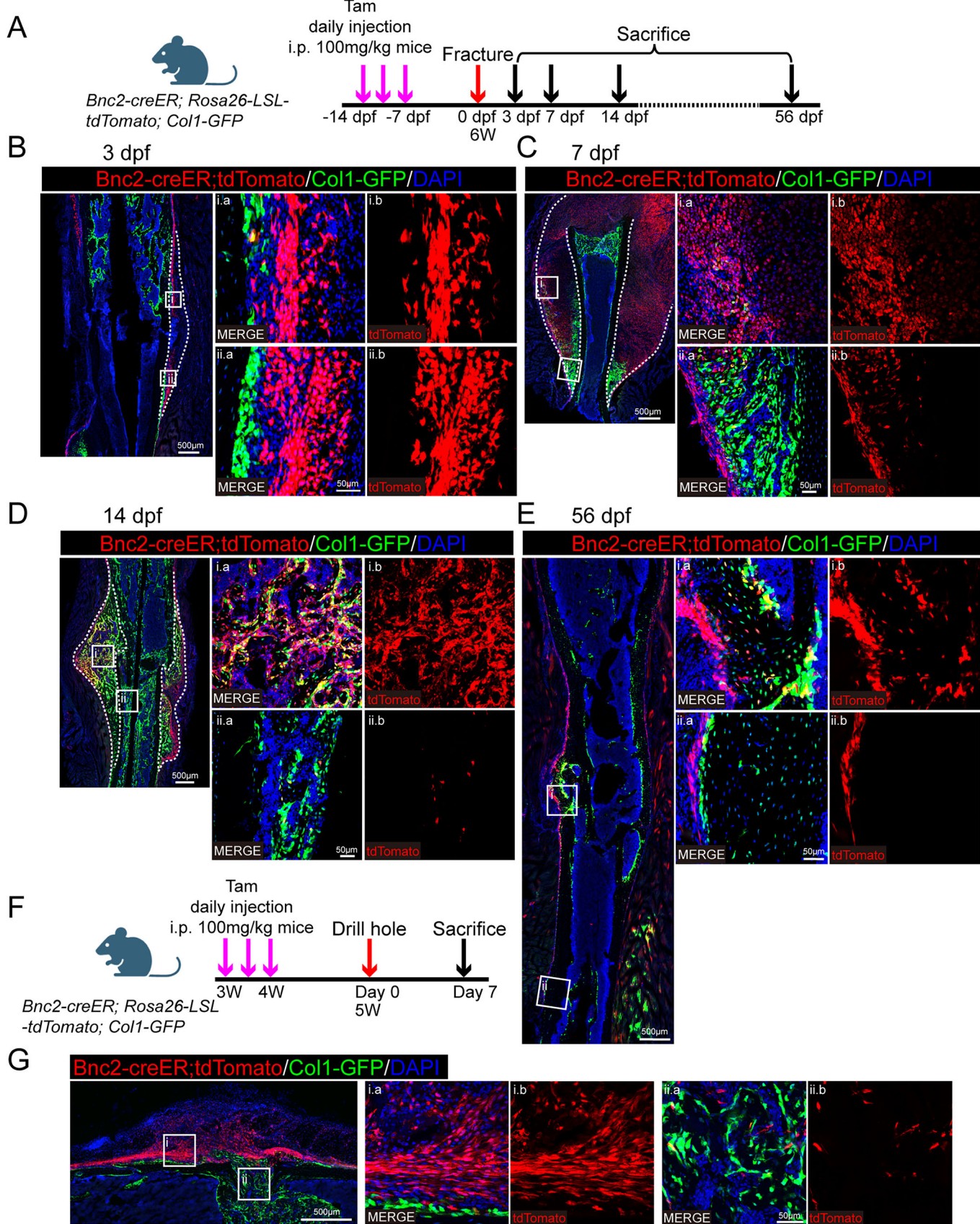

Figure 3. *Bnc2-creER*⁺ periosteal cells are involved in fracture healing.

(A) Schematic diagram of *Bnc2-creER; Rosa26-LSL-tdTomato; Col1-GFP* mouse fracture model. (B) Slides on 3 dpf revealed *Bnc2-creER; Rosa26-LSL-tdTomato* positive cells specifically labeled thickened periosteum. The area within the dotted line is the callus. (C) Slides on 7 dpf revealed *Bnc2-creER; Rosa26-LSL-tdTomato* positive cells labeled a large number of chondrocytes, but significantly fewer osteoblasts which were derived from intramembrane osteogenesis. The area within the dotted line is the callus. (D) Slides on 14 dpf showed *Bnc2-creER; Rosa26-LSL-tdTomato* positive cells labeled a large number of osteoblasts in the external callus and almost no osteoblasts in the internal callus. The area within the dotted line is the callus. (E) Slides on 56 dpf revealed *Bnc2-creER; Rosa26-LSL-tdTomato* positive cells could form new periosteal cells, osteoblasts and osteocytes at the fracture site, while *Bnc2-creER; Rosa26-LSL-tdTomato* positive cells still only labeled periosteal cells at the distal fracture site. (F) Schematic diagram of a bone drilling model performed on *Bnc2-creER; Rosa26-LSL-tdTomato; Col1-GFP* mice. (G) The labeling of *Bnc2-creER; Rosa26-LSL-tdTomato* in periosteum and traumatic region on the 7th day after drilling. Source data are available online for this figure.

periosteal cells decreased significantly (Fig. 6G,H). In addition, a significant increase in EdU⁺BNC2-EGFP⁺ cells was also detected in *Bnc2-EGFP* mice after fracture (Appendix Fig. S8A,B). In addition, periosteal cells were cultured in vitro, and the proliferation ability of these cells was assessed by CCK8 assay and cell cycle analysis. The proliferation capacity of periosteal cells was significantly decreased in *Prx1-cre; Bnc2^{f/f}* mice (Fig. 6I,J). Overexpression of *Bnc2* significantly increased the proliferation capacity of periosteal cells (Appendix Fig. S9A,B). To determine whether BNC2 affects chondrogenesis, we induced the periosteal cells of *Prx1-cre; Bnc2^{f/f}* mice differentiated into chondrocytes. The results showed that loss of BNC2 did not affect the chondrogenic differentiation (Appendix Fig. S9C–E). And depletion of BNC2 with *Prx1-cre* did not show cartilage phenotype in homeostasis (Appendix Fig. S10). However, we detected a significant disorder in cartilage formation in the callus of *Prx1-Cre; Bnc2^{f/f}* mice (Fig. 5H,I). In addition, we found that the absence of *Bnc2* led to a significant decrease in the proliferation ability of periosteal cells (Fig. 6). Therefore, we believe that the cartilage formation disorder during the fracture repair process in *Prx1-Cre; Bnc2^{f/f}* mice is mainly caused by the obstruction of precursor cell proliferation. The above results show that loss of *Bnc2* in periosteal SSCs can lead to significant impaired proliferation in the initial stage of fracture repair, suggesting that *Bnc2* probably regulates the fracture healing process through the activation of periosteal SSCs.

## BNC2 regulates fracture healing by interacting with the NuRD complex

To explore how *Bnc2* regulates the proliferation of SSCs, we used mass spectrometry to identify the BNC2 interacting proteins. Silver staining results showed that Flag-BNC2 was significantly enriched, and specific proteins were pulled down (Fig. 7A). Mass spectrometry analysis showed that the components of the NuRD (nucleosome remodeling and deacetylase) complex were significantly enriched in the Flag-BNC2 immunoprecipitation samples (Fig. 7B). Further co-immunoprecipitation experiments confirmed that there was a significant interaction between BNC2 and the components of the NuRD complex (Fig. 7C). The NuRD complex is one of the major chromatin remodeling complexes, and plays an important role in regulating gene transcription, genome integrity and cell cycle progression (Basta, 2015). The NuRD complex can regulate gene transcription by binding to enhancers and promoters of genes to reduce the level of histone acetylation. Therefore, we use *Prx1-cre; Bnc2^{f/+}; Rosa26-LSL-tdTomato* and *Prx1-cre; Bnc2^{f/f}; Rosa26-LSL-tdTomato* mice to obtain tdTomato-positive cells on 3 dpf by flow cytometry (Appendix Fig. S11A). The acetylation

levels of histone H3 were significantly reduced in tdTomato positive callus cells of *Prx1-cre; Bnc2^{f/f}; Rosa26-LSL-tdTomato* mice (Fig. 7D). Meanwhile, the decreased H3 acetylation in periosteal cells of *Prx1-cre; Bnc2^{f/f}* mice was also demonstrated by pan-H3ac immunohistochemical staining (Fig. 7E). Then, we tried to demonstrate whether the impairment of fracture healing in *Prx1-cre; Bnc2^{f/f}* mice could be alleviated by intervention with the NuRD complex. Through intraperitoneal injection of the HDAC inhibitor TSA in *Prx1-cre; Bnc2^{f/f}* and *Bnc2^{f/f}* mice, we observed that TSA could partially restore the phenotype of fracture healing disorder in *Prx1-cre; Bnc2^{f/f}* mice, as manifested by a partial increase in callus volume (Fig. 7F,G) and increased bone volume (Fig. 7H,I) at the callus on 14 dpf. The callus composition of *Prx1-cre; Bnc2^{f/f}* mice treated by TSA is basically the same as that of *Bnc2^{f/f}* mice, except that the overall volume is still smaller than *Bnc2^{f/f}* mice, as demonstrated by SOFG staining (Fig. 7J).

To further explore the pathways through which BNC2 regulates the proliferation of SSCs in fracture healing, SSCs of *Prx1-cre; Bnc2^{f/f}* and control mice on 3 dpf were obtained by flow cytometry for ATAC-seq. We found a significant decrease in chromatin accessibility near the gene transcription start sites (TSS) of callus SSCs in *Prx1-cre; Bnc2^{f/f}* mice (Fig. 7K). The pathway related to transcriptional activation, cell proliferation and differentiation were significantly enriched through GO analysis of the down-regulated peaks in *Prx1-cre; Bnc2^{f/f}* mice callus SSCs (Fig. 7L). These data suggest that the loss of *Bnc2* leads to a decrease in chromatin accessibility of genes related to transcriptional activation and cell proliferation in callus SSCs, thus hindering the proliferation of callus SSCs.

These results indicate that BNC2 regulates the acetylation levels of histone H3 through its interaction with the NuRD complex, thus affecting chromatin accessibility and ultimately regulating gene transcriptional activation. Pharmacological inhibition of HDAC could partially alleviate the phenotype of fracture healing disorder in *Prx1-cre; Bnc2^{f/f}* mice.

## Discussion

In this study, we found that the expression of *Bnc2* was significantly upregulated in SSCs in the early stages of fracture. By lineage tracing of constructed fluorescent reporter mice, we found that *Bnc2* could label periosteal cells and could be fully involved in fracture repair, ultimately forming neo-bone and periosteum. However, *Bnc2*-labeled periosteal cells do not differentiate into osteoblasts and osteocytes in homeostasis, indicating that they are in a resting state and can be activated and participate in repair only when receiving injury stimulation.

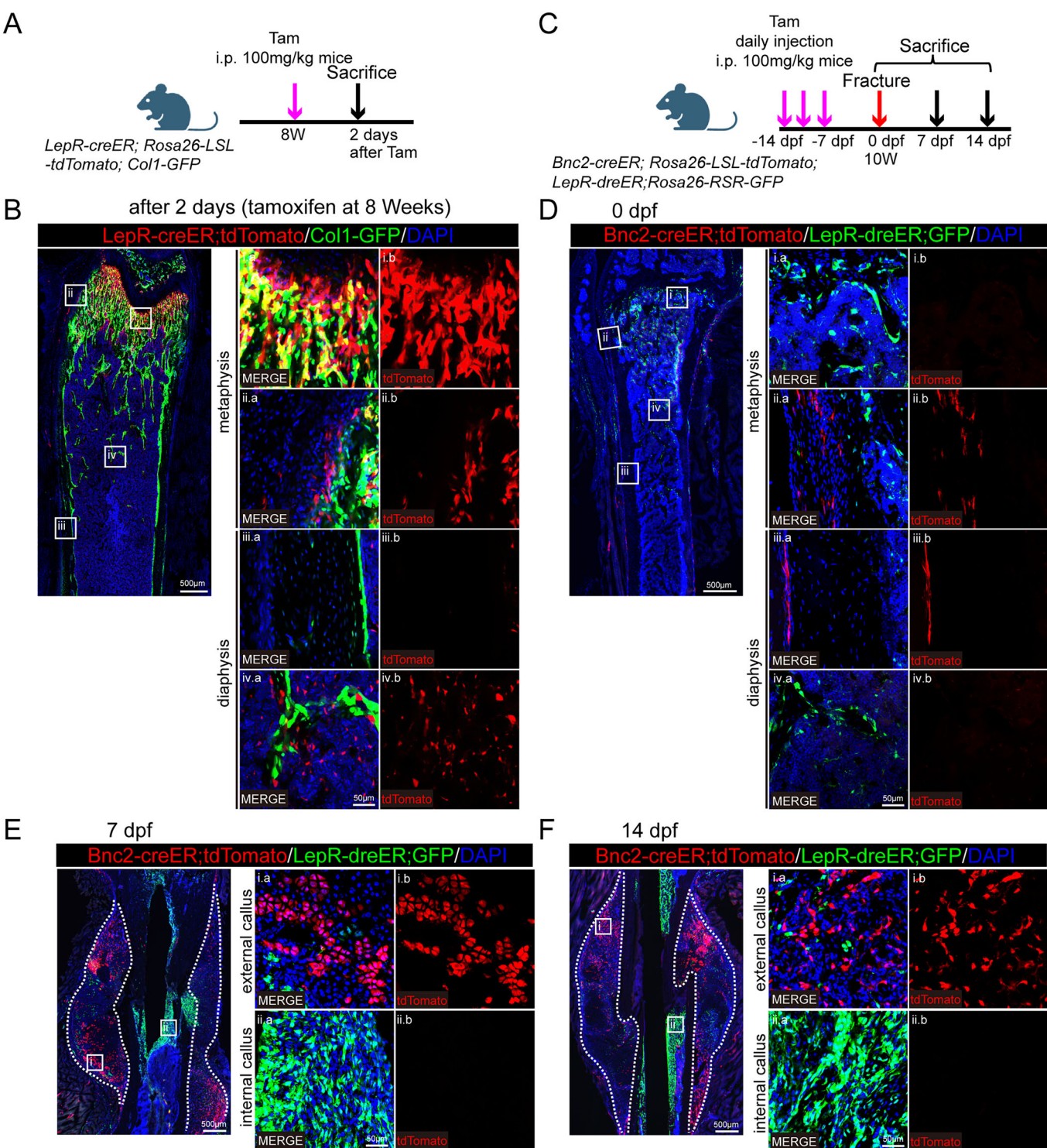

**Figure 4.** ***Bnc2-creER*⁺ periosteal cells are specifically involved in the formation of external callus.**

(**A, B**) Lineage tracing of 8-week-old *LepR-creER; Rosa26-LSL-tdTomato; Col1-GFP* mice at 2 days after tamoxifen induction. (**C**) Schematic diagram of *Bnc2-creER; Rosa26-LSL-tdTomato; LepR-dreER; Rosa26-RSR-GFP* mouse fracture model. (**D**) Lineage tracing of 8-week-old *Bnc2-creER; Rosa26-LSL-tdTomato; LepR-dreER; Rosa26-RSR-GFP* mice at 7 days after tamoxifen induction (0 dpf). (**E, F**) Slides on 7 dpf (**E**) and 14 dpf (**F**) showed *Bnc2-creER; Rosa26-LSL-tdTomato* positive cells participated in the formation of external callus, but not in the formation of internal callus. And *LepR-dreER; Rosa26-RSR-GFP* positive cells were mainly involved in the formation of internal callus but contributed little to the formation of external callus. The area within the dotted line is the callus. Source data are available online for this figure.

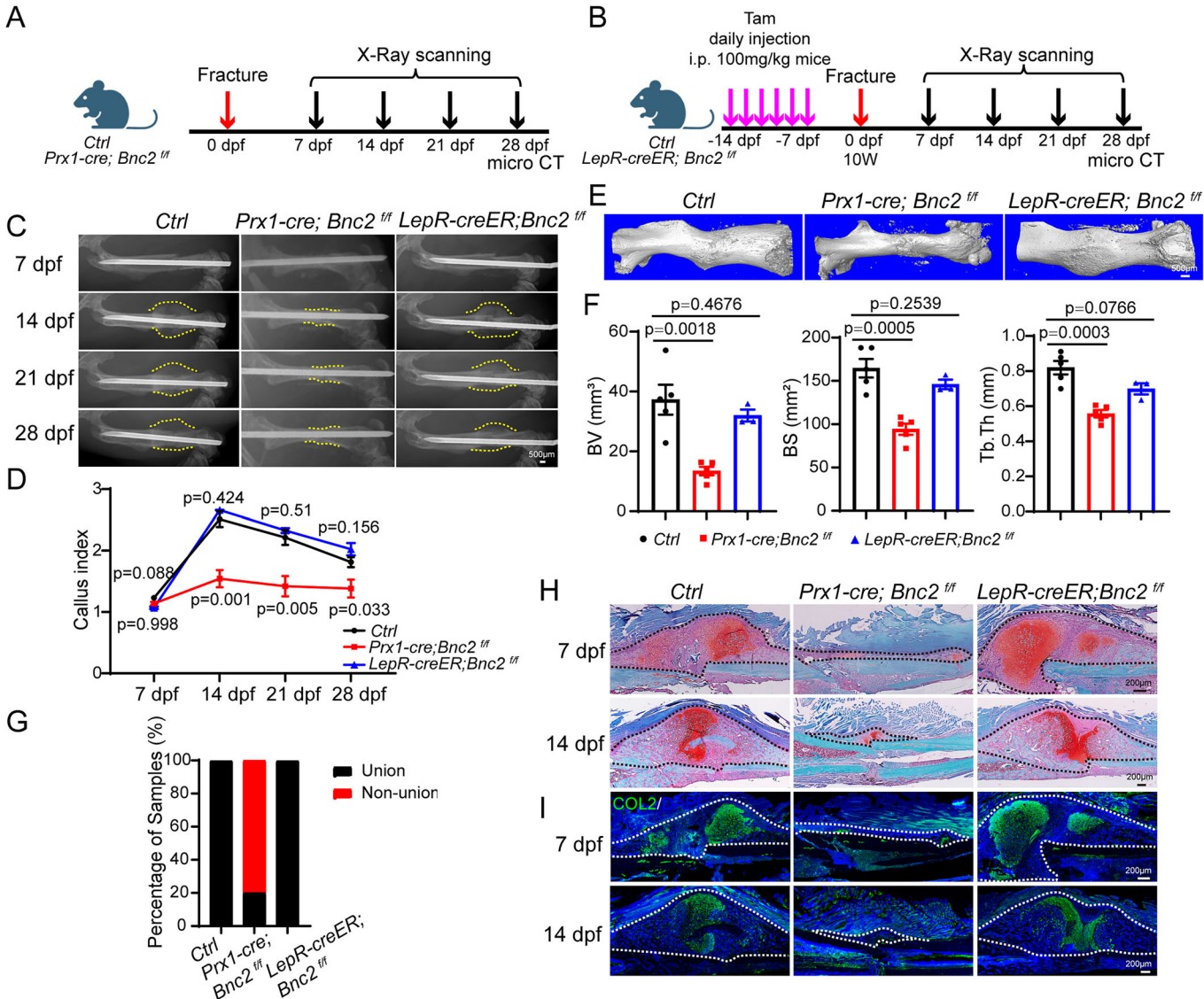

**Figure 5. Loss of *Bnc2* in periosteal SSCs shows significant impairment in fracture healing.**

(A) Schematic diagram of *Prx1-cre; Bnc2^{f/f}* mice fracture model. (B) Schematic diagram of *LepR-creER; Bnc2^{f/f}* mice fracture model. (C, D) X-ray results (C) and quantitative statistics of callus index (D) in *Prx1-cre; Bnc2^{f/f}* and *LepR-creER; Bnc2^{f/f}* mice at different stages after fracture. Data were presented as the means ± SEM. *n* = 5, 5, 3. Unpaired *t*-test. The dotted line represents the edge of the callus. (E, F) Micro-CT scanning results (E) and quantitative statistics of new bone formation at the fracture site (F) of *Prx1-cre; Bnc2^{f/f}* and *LepR-creER; Bnc2^{f/f}* mice on 28 dpf. Data were presented as the means ± SEM. *n* = 5, 5, 3. Unpaired *t*-test. (G) Quantitative statistics of percentage of union and nonunion fracture samples on 28 dpf in *Prx1-cre; Bnc2^{f/f}* and *LepR-creER; Bnc2^{f/f}* mice. *n* = 5, 5, 3. (H) SOFG staining of *Prx1-cre; Bnc2^{f/f}* and *LepR-creER; Bnc2^{f/f}* mice on 7 dpf and 14 dpf. The area within the dotted line is the callus. (I) COL2 immunofluorescence staining of *Prx1-cre; Bnc2^{f/f}* and *LepR-creER; Bnc2^{f/f}* mice on 7 dpf and 14 dpf. The area within the dotted line is the callus. Source data are available online for this figure.

In order to explore how *Bnc2* regulates fracture repair, we constructed a conditional knockout mouse model of *Bnc2*. The fracture repair of *Prx1-cre; Bnc2^{f/f}* mice was significantly abnormal, which was manifested as periosteal cell proliferation disorder and subsequent endochondral osteogenesis obstruction. However, *Ocn-cre; Bnc2^{f/f}* and *LepR-creER; Bnc2^{f/f}* mice did not have the above phenotype, suggesting that *Bnc2* specifically promoting fracture repair in stem/precursor cells of periosteum. Moreover, we found that the chondrogenic differentiation characteristics of periosteal cells of *Prx1-cre; Bnc2^{f/f}* mice were unchanged. This may imply that the loss of *Bnc2* leads to the premature entry of periosteal cells into

the differentiation stage, and the cause of impaired fracture healing in *Prx1-cre; Bnc2^{f/f}* mice may be due to the decreased proliferation ability of precursor cells. Significant downregulation of *BNC2* was detected in both *ZKSCAN3* knockout and *CLOCK* knockout human mesenchymal stem cell senescence models (Hu et al, 2020; Liang et al, 2021). Therefore, we can hypothesize that the deletion of *Bnc2* leads to the premature aging of periosteal cells, which is manifested by a decrease in proliferative ability and early entry into differentiation, which ultimately leads to the lack of sufficient precursor cells in the later stage of fracture repair and eventually causes impaired fracture repair. This study demonstrates that only

*Prx1-cre; Bnc2^{f/f}* mice exhibit an impaired fracture repair phenotype, whereas *LepR-creER; Bnc2^{f/f}* mice complete fracture repair normally. This indicates that *Bnc2* can serve as a factor that specifically regulates the functional involvement of periosteal cells in fracture repair. However, lineage tracing results indicate that *Bnc2* gene primarily labels periosteal cells but also labels a small number of metaphyseal bone marrow cells, suggesting that *Bnc2* is not a perfect marker gene for periosteal cells. More ideal marker genes for periosteal stem cells may yet be identified.

Epigenetic regulation is crucial for bone development and repair. Histone H3K4 tri-methyltransferase *Ash1l* regulates the differentiation of mesenchymal stem cells by epigenetic means (Yin et al, 2019). *Ptip* regulates the proliferation and differentiation of COL2-positive precursor cells by decreasing the H3K27ac level in the *Pgk1* promoter region (Liang et al, 2024). However, the epigenetic control of skeletal stem cells fate remains to be studied. We found that *Bnc2* regulates periosteal cell proliferation by changing chromatin accessibility of transcriptional activators through interaction with the NuRD complex, and thus affects fracture repair, which is actually the regulation of histone acetylation on *Bnc2*-positive periosteal cells from resting to activated state.

We found that *Bnc2*^+ periosteal cells are crucial for fracture repair, but not all *Bnc2*^+ periosteal cells are SSCs. Therefore, by combining with the identified SSC marker gene, we can assume that *Bnc2*^+ SSCs in the periosteum are mainly involved in fracture repair, while *Bnc2*^- SSC is mainly involved in homeostasis maintenance. This is the direction we continue to explore. In addition, we found that BNC2 regulated the expression of genes related to proliferation and differentiation pathways through interaction with the NuRD complex, but the key direct downstream target genes of BNC2 were not identified. And due to the wide range of action of HDAC inhibitors, while promoting fracture repair, HDAC inhibitors may have temporarily unknown side effects. Subsequently, we will further identify the target genes of BNC2 to achieve more precise regulation and provide new targets for the treatment of nonunion of bone fractures.

In summary, we found that *Bnc2* specifically labeled resting periosteal cells in homeostasis, and that *Bnc2*-positive periosteal cells contributed adequately to endochondral osteogenesis during fracture repair. Meanwhile, BNC2 regulates the proliferation of periosteal SSCs in an epigenetic manner through interaction with the NuRD complex in the initial stage of fracture repair (Appendix Fig. S12A). This study suggests that intervention in histone acetylation may promote the recovery of some bone repair defects caused by stem cell proliferation disorders.

# Methods

### Reagents and tools table

| Reagent/resource | Reference or source | Identifier or catalog number |
|---|---|---|
| **Experimental models** | | |
| *Bnc2-P2A-EGFP* mice | GemPharmatech | Customized |
| *Bnc2-P2A-creER* mice | GemPharmatech | Customized |

| Reagent/resource | Reference or source | Identifier or catalog number |
|---|---|---|
| *Bnc2^{f/f}* mice | GemPharmatech | Customized |
| *Ocn-creER* mice | BIOCYTOGEN | Customized |
| *Prx1-Cre* mice | Jackson Laboratory | B6.Cg-Tg(Prrx1-cre)1Cjt/J |
| *Rosa26-LSL-tdTomato* mice | Zilong Qiu (Shanghai Jiao Tong University, Shanghai, China) | |
| *LepR-dreER; Rosa26-RSR-GFP* mice | Bo O. Zhou (CAS Center for Excellence in Molecular Cell Science, Shanghai, China) | |
| *LepR-creER* mice | Bo O. Zhou (CAS Center for Excellence in Molecular Cell Science, Shanghai, China) | |
| *Ocn-cre* mice | Peiqiang Su (The First Affiliated Hospital, Sun Yat-sen University, Guangzhou, China) | |
| *Prx1-creER* mice | Baojie Li (Shanghai Jiao Tong University) | |
| **Recombinant DNA** | | |
| Phage-Flag-Bnc2 | This study | |
| Plenti-HA-Bnc2 | This study | |
| Plex-Flag-Mta1 | This study | |
| Plex-Flag-Mta2 | This study | |
| Plex-Flag-Rbbp4 | This study | |
| Plex-Flag-Rbbp7 | This study | |
| Plex-Flag-Gatad2a | This study | |
| Plex-Flag-Gatad2b | This study | |
| Plex-Flag-Mbd3 | This study | |
| Plex-Flag-Hdac1 | This study | |
| **Antibodies** | | |
| Rabbit anti-COL2 | Boster | Cat # BA0533 |
| Goat anti-OPN | R&D | Cat # AF808 |
| Rabbit anti-pan-H3ac | Abcam | Cat # ab300641 |
| Rabbit anti-H3 | Cell Signaling Technology | Cat #9715 |
| Mouse anti-Flag | Sigma | Cat #F3165 |
| Rabbit anti-HA | Cell Signaling Technology | Cat #3724 |
| Donkey anti-rabbit 488 | Molecular Probes | Cat # A21206 |
| Donkey anti-goat Cy3 | Jackson ImmunoResearch | Cat #705-165-147 |
| PerCP/Cy5.5-conjugated anti-CD45 | Biolegend | Cat #103132 |
| PerCP/Cy5.5-conjugated anti-CD31 | Biolegend | Cat #102420 |
| PerCP/Cy5.5-conjugated anti-Ter119 | Biolegend | Cat #116228 |
| FITC-conjugated anti-mouse 6C3/Ly-51 | Biolegend | Cat #108305 |

| Reagent/resource | Reference or source | Identifier or catalog number |
|---|---|---|
| Brilliant Violet 605™-conjugated anti-mouse CD90.2 | Biolegend | Cat #140317 |
| PE/Cy7-conjugated anti-mouse CD105 | Biolegend | Cat #120409 |
| APC-conjugated anti-mouse CD200 | Biolegend | Cat #123809 |
| **Oligonucleotides and other sequence-based reagents** | | |
| PCR Primers | This study | |
| **Chemicals, enzymes and other reagents** | | |
| Trichostatin A | MCE | Cat # HY-15144 |
| FLAG-M2 peptide | Sigma | Cat #F3290 |
| Protease inhibitors | MCE | Cat # HY-K0010 |
| Protein A/G PLUS-Agarose | Santa Cruz | Cat # sc-2003 |
| Flag-M2 agarose beads | Sigma | Cat # A2220 |
| Silver staining kit | Sangon Biotech | Cat # C510027 |
| Red blood cell lysis buffer | Beyotime | Cat #C3702 |
| α-MEM | Corning | Cat #10-022-CVR |
| Collagenase | Sigma | Cat # C0130 |
| Dispase II | Sigma | Cat # D4693 |
| EdU | RiboBio | Cat # C00054 |
| Cell-Light EdU Apollo643 In Vitro Kit | RiboBio | Cat # C10310-2 |
| Cell-Light EdU Apollo488 In Vitro Kit | RiboBio | Cat # C10310-3 |
| DAPI | Sigma | Cat #D9542 |
| Fluorescence mounting medium | DAKO | Cat # S3023 |
| Tamoxifen | Sigma | Cat # T5648 |
| OCT-freeze medium | Epredia | Cat #22-110-6502 |
| **Software** | | |
| Flowjo v10.8.1 | BD | |
| GraphPad Prism v8 | GraphPad | |
| Illustrator | Adobe | |
| **Other** | | |
| VS-120 microscope | Olympus | |
| Sp8 STED microscope | Leica | |
| LSM980 microscope | ZEISS | |
| Cytoflex | Beckman Coulter | |
| BD Arial Fusion | BD | |
| MX2 X-ray system | Vet Ray Technology | |
| SkyScan 1272 | Bruker | |

## Animals

*Bnc2-P2A-EGFP*, *Bnc2-P2A-creER,* and *Bnc2^{f/f}* mice were generated with the assistance of GemPharmatech. *Ocn-creER* mice were generated with the assistance of BIOCYTOGEN. *Prx1-Cre* mice was purchased from the Jackson Laboratory. *Rosa26-LSL-tdTomato* mice were kindly provided by Dr. Zilong Qiu. *LepR-dreER; Rosa26-RSR-GFP* and *LepR-creER* mice were kindly provided by Dr. Bo O. Zhou. *Ocn-cre* mice were kindly provided by Dr. Peiqiang Su. *Prx1-creER* mice were kindly provided by Dr. Baojie Li. CreER expressing mice were induced by intraperitoneal injection daily with 100 mg/kg mice of tamoxifen dissolved in corn oil for three or six times. All utilized mice were maintained on the C57/BL6 background, both male and female mice were analyzed. All animals were housed under specific pathogen-free conditions.

## Mouse femoral fracture model

Six to ten-week-old mice were used for fracture modeling. Mice were anesthetized with Avertin. Intramedullary fixation was performed on the femur with a pin, and the fracture was performed in the middle of the femur with a dentist's microdrill. After the muscle was repositioned, absorbable sutures were used to close the skin. X-ray were performed weekly after the fracture with the MX2 X-ray system, and micro-CT scans were performed 4 or 5 weeks after surgery with Bruker, SkyScan 1272. Callus index was defined as the ratio of the maximum diameter of the callus to the diameter of the femur.

## Mouse bone drill hole model

The drill hole model is used to characterize the bone injury repair model, mainly involved in the intramembrane osteogenesis (Colnot, 2009). Briefly, a skin incision was made in the middle of the tibia, the subcutaneous tissue was bluntly separated, the tibia was exposed, and then a 0.7 mm diameter needle was used to drill through one side of the cortical bone. Once this was done, the skin was repositioned and sutured. Bone formation was detected by micro-CT on the 14th day after surgery.

## Histological analysis

Freshly isolated mouse samples were fixed at 4 °C with 4% PFA for 48 h, and then decalcified at 350 mM EDTA, PH6.5 after washing with PBS. For paraffin sections, the samples were dehydrated by concentration gradient ethanol and then paraffin-embedded and sliced at a thickness of 7 μm. For frozen sections, samples were dehydrated with 30% sucrose and then embedded with tissue OCT-freeze medium (Epredia, 22-110-6502) and sliced at a thickness of 20 μm.

## Immunohistochemistry and immunofluorescence staining

The frozen sections were restored to room temperature and rehydrated with PBS. Then the antigenic repair was performed with protease K in a water bath at 37 °C for 10 min. The sections were blocked and permeabilized with PBS containing 10% horse serum and 0.3% Triton X-100 for 1 hour at room temperature. The

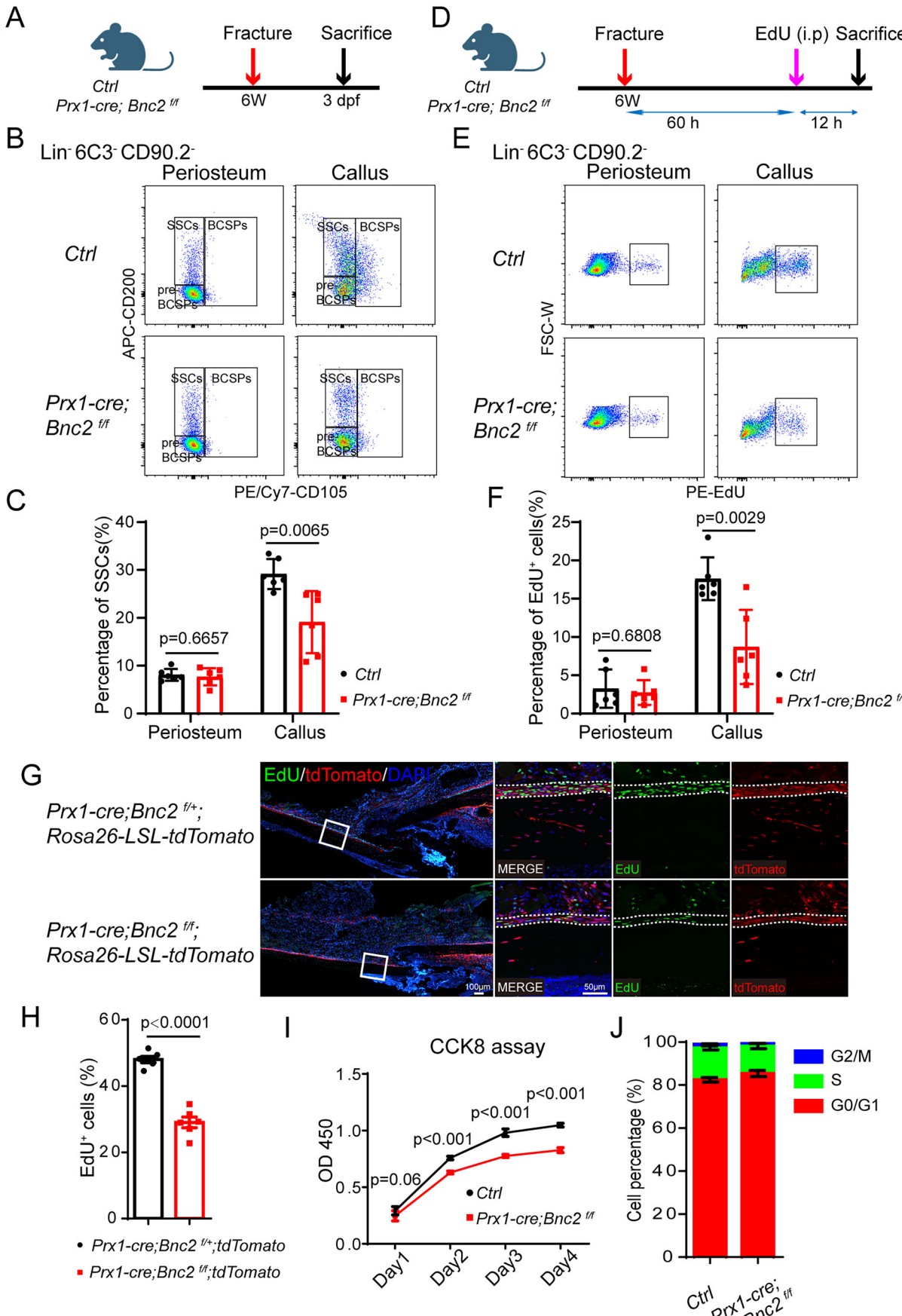

◄  **Figure 6.  *Prx1-cre; Bnc2^{f/f}* mice have a significant decrease in SSC activation in the early stages of fracture healing.**

(A) Schematic diagram of SSCs analysis in *Prx1-cre; Bnc2^{f/f}* mice after fracture. (B, C) Flow analysis of the proportion of SSCs in undamaged periosteum and 3 dpf callus in *Prx1-cre; Bnc2^{f/f}* mice. Data were presented as the means ± SEM. $n = 6$. Unpaired *t*-test. (D) Schematic diagram of in vivo EdU labeling in *Prx1-cre; Bnc2^{f/f}* mice after fracture. (E, F) Flow analysis of the proportion of EdU-positive precursor cells in undamaged periosteum and 3 dpf callus in *Prx1-cre; Bnc2^{f/f}* mice. Data were presented as the means ± SEM. $n = 6$. Unpaired *t*-test. (G, H) EdU staining and quantitative statistics of the proportion of EdU-positive periosteal cells in 2 dpf sections of *Prx1-cre; Bnc2^{f/f}* mice. Data were presented as the means ± SEM. $n = 6$. Unpaired *t*-test. (I) CCK8 assay of periosteal cells in *Prx1-cre; Bnc2^{f/f}* mice. Data were presented as the means ± SEM. $n = 6$. Unpaired *t*-test. (J) Cell cycle analysis of periosteal cells in *Prx1-cre; Bnc2^{f/f}* mice. Data were presented as the means ± SEM. $n = 3$. Unpaired *t*-test. Source data are available online for this figure.

primary antibody (rabbit anti-COL2, Boster, BA0533, 1:200; goat anti-OPN, R&D, AF808, 1:500; rabbit anti-pan-H3ac, Abcam, ab300641, 1:200) was incubated at 4 °C overnight and washed with PBS. The secondary antibody coupled with fluorescein (donkey anti-rabbit 488, Molecular Probes, A21206, 1:1000; donkey anti-goat Cy3, Jackson ImmunoResearch, 705-165-147, 1:1000) was incubated at room temperature for 1 hour and washed with PBS. The nucleus was counterstained with DAPI (Sigma, D9542) and finally mounted with fluorescence mounting medium (Dako, S3023). Stained images were acquired with Olympus VS-120, Leica Sp8 STED and ZEISS LSM980 microscopes.

## In vivo EdU labeling and staining

EdU solution (5 mg/mL) (RiboBio, C00054) was intraperitoneally injected into mice with 10 μL/g of mice. After 12 h of injection, the mice were sacrificed, and the hind limbs were collected. The section sample was prepared according to the conventional paraffin section preparation method. The EdU staining procedure was performed according to the kit's instructions (RiboBio, C10310-2 and C10310-3).

## Isolation of periosteal cells

The muscles of the femur and tibia were removed first, then both ends of the bone were coated with a low melting agarose (10% in TAE buffer) and digested with minimum essential medium alpha (α-MEM; Corning, 10-022-CVR) containing 1 mg/mL collagenase (Sigma, C0130) and 2 mg/mL Dispase II (Sigma, D4693) at 37 °C, 150 rpm, 30 min × 2. The cell suspension can be filtered through a 70-μm cell strainer (Falcon, 352350) for flow cytometry, sorting, or induction of differentiation.

## Flow cytometry

In brief, freshly isolated primary cells were treated with red blood cell lysis buffer (Beyotime, C3702) and then stained with antibodies as following: PerCP/Cy5.5-conjugated anti-CD45 (Biolegend, 103132), PerCP/Cy5.5-conjugated anti-CD31 (Biolegend, 102420), PerCP/Cy5.5-conjugated anti-Ter119 (Biolegend, 116228), FITC-conjugated anti-mouse 6C3/Ly-51 (Biolegend, 108305), Brilliant Violet 605™-conjugated anti-mouse CD90.2 (Biolegend, 140317), PE/Cy7-conjugated anti-mouse CD105 (Biolegend, 120409), APC-conjugated anti-mouse CD200 (Biolegend, 123809). The above primary antibody was diluted by 1:500 and incubated on ice for 30 min away from light. After washing with 2% FBS PBS, flow cytometry was performed with Cytoflex (Beckman Coulter) and cell sorting was performed with BD Arial Fusion (BD Biosciences), and data were analyzed with FlowJo software.

## Flag-BNC2 pull-down

Flag-BNC2 plasmid was transfected into 293FT cells for the overexpression of the Flag-BNC2 protein. Two days after transfection, cells were collected and treated with EBC buffer containing protease inhibitors (MCE, HY-K0010). Then, protein A/G PLUS-Agarose (Santa Cruz, sc-2003) was added to the cell lysate and shaken at 4 °C for 3 h for preincubation, and the supernatant was transferred after centrifugation. Added Flag-M2 agarose beads (Sigma, A2220) and incubated overnight at 4 °C. Discarded the supernatant after centrifuging, rinsed with pre-cooled EBC three times, then eluted protein with FLAG-M2 peptide (100 μg/mL) (Sigma, F3290). After SDS-PAGE electrophoresis, the samples were silver-stained to detect protein enrichment (Sangon Biotech, C510027). Finally, the eluted liquid samples were identified by LC-MS/MS.

## Pharmacological inhibition of HDACs in mice

*Bnc2^{f/f}* and *Prx1-cre; Bnc2^{f/f}* mice were randomly grouped and intraperitoneally injected daily with TSA (MCE, HY-15144) or solvent as controls at 10 mg/kg starting on day 1 after fracture and continuing until day 14 after surgery according to Amy M Avila's research(Avila et al, 2007). The mice were sacrificed at 14 dpf, and the fracture repair status was examined by X-ray, micro-CT, and histological analysis.

## Micromass culture

Periosteal cells were digested, resuspended at $1 \times 10^7$ cells/mL, and plated in a 12.5 μL droplet of cell suspension in the center of a 24-well-plate; the plate was placed at 37 °C for 2 h, and chondrogenic differentiation medium, which contained DMEM (Corning, 10-013-CVR), 10 ng/mL TGFβ3 (Peprotech, 100-36E), 100 nM dexamethasone (Sigma, D1756), 50 μg/mL L-ascorbic acid 2-phosphate (Sigma, A8960), 1 mM sodium pyruvate (Sigma, 25-000-CIR), 40 μg/mL proline (Sigma, P5607), and 1% ITS (Cyagen, ITSS-10201-10), was then gently added. On the fourth day of induced differentiation, micromasses were acidified with 0.1 N HCl and were then stained with 1% Alcian blue (Sigma, A5268) or RNA extraction to detect the expression of marker genes.

## Quantitative real-time PCR

Total RNA was isolated from different tissues and cells with TRIzol Reagent (Sigma, T9424) and reverse-transcribed with a PrimeScript RT Reagent Kit (Takara, RR037A). Real-time fluorescence quantitative PCR was performed in a Bio-Rad CFX Connect Real-Time System. The primer sets used were *Bnc2*: sense 5'-GTGATCAGTGTAAACATGGCTGGGT-3',

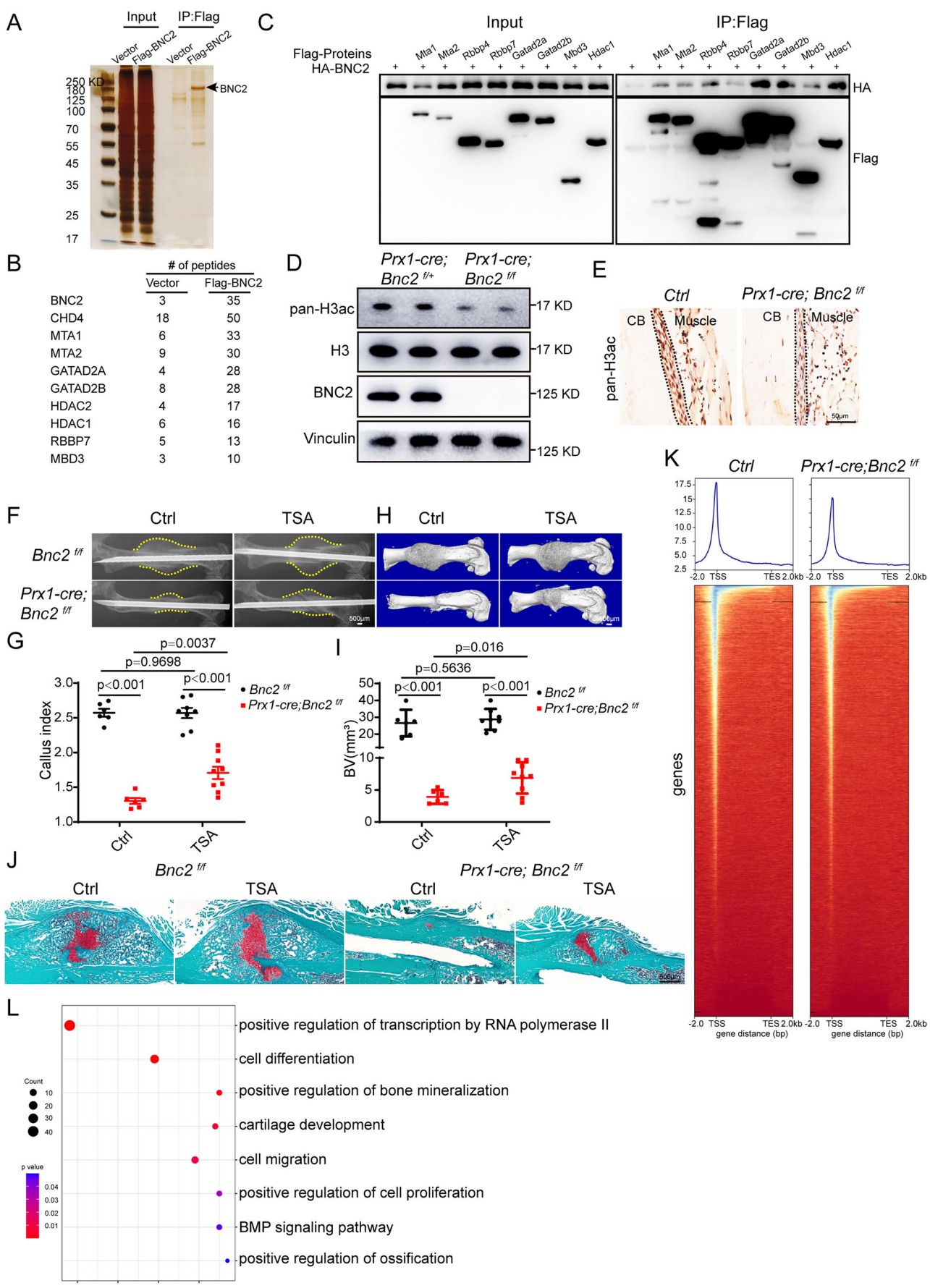

◄  **Figure 7.  BNC2 regulates H3ac of callus cells through the NuRD complex.**

(A) Silver staining results of Flag-BNC2 pull-down. (B) Enrichment of components of the NuRD complex in LC-MS/MS results of Flag-BNC2 pull-down. (C) Co-immunoprecipitation to verify the interaction between BNC2 and components of the NuRD complex. (D) Pan-H3ac was detected by WB in Lin⁻ *Prx1-cre; Rosa26-LSL-tdTomato*⁺ cells of *Prx1-cre; Bnc2*^f/f^ mice in 3 dpf. (E) Pan-H3ac levels in periosteal cells of *Prx1-cre; Bnc2*^f/f^ mice on 2 dpf were determined by immunohistochemistry. CB, cortical bone. (F, G) X-Ray results (F) and callus index (G) on 14 dpf in TSA-treated *Prx1-cre; Bnc2*^f/f^ mice. Data were presented as the means ± SEM. $n = 6, 6, 8, 9$. Unpaired *t*-test. The dotted line represents the edge of the callus. (H, I) Micro-CT scanning results (H) and bone volume (I) at callus of 14 dpf in TSA-treated *Prx1-cre; Bnc2*^f/f^ mice. Data were presented as the means ± SEM. $n = 6, 6, 8, 9$. Unpaired *t*-test. (J) SOFG staining of sections on 14 dpf in TSA-treated *Prx1-cre; Bnc2*^f/f^ mice. (K) ATAC-seq results of SSCs on 3 dpf in *Prx1-cre; Bnc2*^f/f^ mice showed chromatin accessibility on the gene body and heat maps of peaks. (L) GO analysis of down-regulated peaks in the SSCs on 3 dpf of *Prx1-cre; Bnc2*^f/f^ mice. Fisher's exact test. Source data are available online for this figure.

*anti-sense* 5'-GGATGTATCCTCGGACATAGTCCCT-3'; *Sox9*: *sense* 5'-TTCCTCCTCCCGGCATGAGTG-3', *anti-sense* 5'- CAACTTTGCCAGCTTGCACG-3'; *Col2*: *sense* 5'- CGGTCCTACGGTGTCAGG-3', *anti-sense* 5'-GCAGAGGACATTCCCAGTGT-3'; *Acan*: *sense* 5'- AATCCCCAAATCCCTCATAC-3', *anti-sense* 5'- CTTAGTCCACCCCTCCT-CAC-3'; *Col10*: *sense* 5'-TTCTGCTGCTAATGTTCTTGACC-3', *anti-sense* 5'- GGGATGAAGTATTGTGTCTTGGG-3'; *Hprt*: *sense* 5'-GTTAAGCAGTACAGCCCCAAA-3', *anti-sense* 5'- AGGGCATATCCAA-CAACAAACTT-3'.

## Bulk RNAseq

RNA purification, reverse transcription, library construction and sequencing were performed at Shanghai Majorbio Bio-pharm Biotechnology Co., Ltd. (Shanghai, China) according to the manufacturer's instructions. The transcriptome library was prepared following Illumina® Stranded mRNA Prep, Ligation (San Diego, CA) using 1 µg of total RNA. After quantification by Qubit 4.0, the sequencing library was performed on the NovaSeq X Plus platform (PE150) using NovaSeq Reagent Kit (NovaSeq 6000). The raw paired-end reads were trimmed and quality-controlled by fastp with default parameters. Then, clean reads were separately aligned to the reference genome with orientation mode using HISAT2 software. The mapped reads of each sample were assembled by StringTie in a reference-based approach.

## ATAC-seq

The cells were then thawed in a 37 °C water bath, pelleted, and washed with cold PBS. The cell pellets were resuspended in lysis buffer, pelleted, and tagmented using the Assembled Tn5 Transposome and Tagmentation buffers (Active Motif). Tagmented DNA was then purified using the QIAGEN PCR Purification Kit, amplified with ten cycles of PCR, and purified using the QIAGEN PCR Purification Kit. The resulting material was quantified with Qubit (Thermo Fisher) and sequenced with PE150 sequencing on the Illumina platform. Raw reads were filtered to obtain high-quality clean reads by removing sequencing adapters, short reads (length <35 bp), and low-quality reads using trim-galore (v0.6.4). Then FastQC (v0.11.9) and Multiqc (v1.8) is used to ensure high reads quality. The clean reads were mapped to the mouse genome (assembly mouse genome mm10) using the Burrow-Wheeler Aligner (BWA v0.7.17) software. PCR duplicates were removed using Picard (v2.22.2-0). Peak detection was performed using the MACS2 (v2.2.6) peak finding algorithm with 0.05 set as the *q* value cutoff and −100 set as the shift. Annotation of peak sites to gene features was performed using the Homer annotatePeaks.pl (v4.10).

TSS heatmap was generated using deepTools (v3.4.3) computeMatrix reference-point.

## Statistics

Data were derived from independently obtained datasets and expressed as mean ± SEM. Two groups were compared using paired or unpaired two-tailed *t*-tests. The number of samples shown in each figure is the number of biological replicates.

## Ethics statement

We comply with all ethical regulations regarding animal testing and research. All animal experiments were conducted in the Animal Laboratory of Shanghai Institute of Biochemistry and Cell Biology, CAS Center for Excellence in Molecular Cell Science, in accordance with the protocol approved by the Committee for Animal Health and Use of Shanghai Institute of Biochemistry and Cell Biology, CAS Center for Excellence in Molecular Cell Science, Chinese Academy of Sciences (Approval number: SIBCB-S350-2403-16).

# Data availability

The datasets produced in this study are available in the following databases: ATAC-Seq data: China National Center for Bioinformation, Genome Sequence Archive (CRA029252, https://ngdc.cncb.ac.cn/gsa/browse/CRA029252). Bulk RNA-Seq data: China National Center for Bioinformation, Genome Sequence Archive (CRA030423, https://ngdc.cncb.ac.cn/gsa/browse/CRA030423).

The source data of this paper are collected in the following database record: biostudies:S-SCDT-10_1038-S44318-025-00664-1.

# Peer review information

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

## Acknowledgements

We thank Zilong Qiu (Shanghai Jiao Tong University, Shanghai, China) for the *Ros26-LSL-tdTomato* mice, Bo O. Zhou (CAS Center for Excellence in Molecular Cell Science, Shanghai, China) for the *LepR-dreER; Rosa26-RSR-GFP* and *LepR-creER* mice, Peiqiang Su (The First Affiliated Hospital, Sun Yat-sen University, Guangzhou, China) for the *Ocn-cre* mice; and Baojie Li (Shanghai Jiao Tong University) for the *Prx1-creER* mice. We also thank the core facility for cell analysis technology and the animal core facility of the CAS Center for Excellence in Molecular Cell Science, Shanghai, China and the core facility for microscopic imaging technology and the animal core facility of the Hainan Academy of Medical Sciences, Hainan, China. The work was supported by the National Natural Science Foundation of China (82272608, 82530082, 82230082, 82102554, and 2502903), the Space Medical Experiment Project of China Manned Space Program (HYZHXM01025), and the Strategic Priority Research Program of the Chinese Academy of Science (grant XDB0570000).

## Author contributions

**Zhong Zhang**: Data curation; Formal analysis; Funding acquisition; Validation; Investigation; Visualization; Methodology; Writing—original draft; Project administration; Writing—review and editing. **Lingli Zhang**: Funding acquisition; Investigation; Methodology; Writing—review and editing. **Bo Jiang**: Data curation; Investigation; Methodology; Writing—review and editing. **Shuqin Chen**: Data curation; Software. **Wenhui Xing**: Investigation. **Peilong Wang**: Investigation. **Lixiang Lou**: Investigation. **Chunxiao Tang**: Investigation. **Xuye Hu**: Investigation. **Jinlong Suo**: Investigation. **Bo O Zhou**: Writing—review and editing. **Weiguo Zou**: Conceptualization; Resources; Supervision; Funding acquisition; Project administration; Writing—review and editing. **Lijun Wang**: Conceptualization; Supervision; Writing—review and editing.

Source data underlying figure panels in this paper may have individual authorship assigned. Where available, figure panel/source data authorship is listed in the following database record: biostudies:S-SCDT-10_1038-S44318-025-00664-1.

## Disclosure and competing interests statement

The authors declare no competing interests.

