## [Peer Review File · The EMBO Journal]

Basonuclin-2 promotes fracture repair through NuRD-dependent chromatin remodeling in periosteal stem cells

Zhong Zhang, Lingli Zhang, Bo Jiang, Shuqin Chen, Wenhui Xing, Peilong Wang, Lixiang Lou, Chunxiao Tang, Xuye Hu, Jinlong Suo, Bo Zhou, Weiguo Zou, and Lijun Wang

Corresponding authors: Lijun Wang (wanglijun2014@sibcb.ac.cn) , Weiguo Zou (zouwg94@sibcb.ac.cn), Lijun Wang (wanglijun2014@sibcb.ac.cn)

Review Timeline:

Submission Date:	9th Apr 25
Editorial Decision:	30th Apr 25
Revision Received:	29th Aug 25
Editorial Decision:	22nd Sep 25
Revision Received:	28th Sep 25
Accepted:	7th Nov 25

Editor: Daniel Klimmeck

Transaction Report:

Dear Dr Zou,

Thank you again for the submission of your manuscript (EMBOJ-2025-121039) to The EMBO Journal. Your manuscript was sent to two reviewers with expertise in skeletal biology and stem cells, and we have now received reports from the both of them, which I enclose below. Based on all information at hand, we have now decided to invite you to revise your work for the EMBO Journal.

As you will see from the experts' reports, the referees acknowledge the analysis and potential interest and value of your findings. However, they also express important issues, which need to be addressed thoroughly to make them supportive of publication in the EMBO Journal. Further, the reviewers raise a number of issues related to the presentation of the findings, statistics applied and overall discussion of related literature, that would need to be conclusively addressed to achieve the level of robustness and clarity needed for The EMBO Journal.

Given the overall interest stated and broader angle of your findings, we are able to invite you to revise your manuscript experimentally to address the referees' comments. I need to stress though that we do require strong support from the referees on a revised version of the study in order to move on to publication of the work.

I would appreciate if you could contact me during the next weeks for exchange e.g. a video call to discuss your perspective on the comments and potential plan for revisions.

Please feel free to contact me if you have any questions or need further input on the referee comments.

When submitting your revised manuscript, please carefully review the instructions below.

Please feel free to approach me any time should you have additional questions related to this.

Thank you for the opportunity to consider your work for publication.

I look forward to your revision.

Kind regards,

Daniel Klimmeck

Daniel Klimmeck, PhD
Senior Editor
The EMBO Journal

Instruction for the preparation of your revised manuscript:

- 1) a .docx formatted version of the manuscript text (including legends for main figures, EV figures and tables). Please make sure that the changes are highlighted to be clearly visible.
- 2) individual production quality figure files as .eps, .tif, .jpg (one file per figure).
- 3) a .docx formatted letter INCLUDING the reviewers' reports and your detailed point-by-point response to their comments. As part of the EMBO Press transparent editorial process, the point-by-point response is part of the Review Process File (RPF), which will be published alongside your paper.
- 4) a complete author checklist, which you can download from our author guidelines ([https://wol-prod-cdn.literatumonline.com/pb-assets/embo-site/Author Checklist%20-%20EMBO%20J-1561436015657.xlsx](https://wol-prod-cdn.literatumonline.com/pb-assets/embo-site/Author%20Checklist%20-%20EMBO%20J-1561436015657.xlsx)). Please insert information in the checklist that is also reflected in the manuscript. The completed author checklist will also be part of the RPF.

6) It is mandatory to include a 'Data Availability' section after the Materials and Methods. Before submitting your revision, primary datasets produced in this study need to be deposited in an appropriate public database, and the accession numbers and database listed under 'Data Availability'. Please remember to provide a reviewer password if the datasets are not yet public (see <https://www.embopress.org/page/journal/14602075/authorguide#datadeposition>).

7) Our journal encourages inclusion of *data citations in the reference list* to directly cite datasets that were re-used and obtained from public databases. Data citations in the article text are distinct from normal bibliographical citations and should directly link to the database records from which the data can be accessed. In the main text, data citations are formatted as follows: "Data ref: Smith et al, 2001" or "Data ref: NCBI Sequence Read Archive PRJNA342805, 2017". In the Reference list, data citations must be labeled with "[DATASET]". A data reference must provide the database name, accession number/identifiers and a resolvable link to the landing page from which the data can be accessed at the end of the reference. Further instructions are available at .

8) At EMBO Press we ask authors to provide source data for the main and EV figures. Our source data coordinator will contact you to discuss which figure panels we would need source data for and will also provide you with helpful tips on how to upload and organize the files.

Numerical data can be provided as individual .xls or .csv files (including a tab describing the data). For 'blots' or microscopy, uncropped images should be submitted (using a zip archive or a single pdf per main figure if multiple images need to be supplied for one panel). Additional information on source data and instruction on how to label the files are available at .

9) We replaced Supplementary Information with Expanded View (EV) Figures and Tables that are collapsible/expandable online (see examples in <https://www.embopress.org/doi/10.15252/embj.201695874>). A maximum of 5 EV Figures can be typeset. EV Figures should be cited as 'Figure EV1, Figure EV2" etc. in the text and their respective legends should be included in the main text after the legends of regular figures.

11) For data quantification: please specify the name of the statistical test used to generate error bars and P values, the number (n) of independent experiments (specify technical or biological replicates) underlying each data point and the test used to calculate p-values in each figure legend. The figure legends should contain a basic description of n, P and the test applied. Graphs must include a description of the bars and the error bars (s.d., s.e.m.).

We realize that it is difficult to revise to a specific deadline. In the interest of protecting the conceptual advance provided by the work, we recommend a revision within 3 months (29th Jul 2025). Please discuss the revision progress ahead of this time with the editor if you require more time to complete the revisions.

Referee #1:

In this study, Zhang et al. systematically explored the role of BNC2 in fracture healing. They identified BNC2+ cells as quiescent periosteal cells under homeostatic conditions, which undergo rapid expansion to promote endochondral ossification and fracture repair without affecting intramembranous ossification. Mechanistically, they found that BNC2 increases the acetylation levels of histone H3 through its interaction with the NuRD complex, thus promoting chromatin accessibility and transcriptional activation. Overall, this is an interesting study that uncovers a critical role of BNC2+ periosteal cells during fracture healing. To further support the authors' conclusions and strengthen the manuscript, there are a few concerns that need to be addressed in a potential revision:

Major points:

1. There is no sufficient ex vivo and in vivo data to support that BNC2 specifically marks periosteal stem cells. Therefore, better describe them as Bnc2+ periosteal cells throughout the manuscript, especially in the title, to avoid controversy. In Figures 1G and 1H, although the percentage of SSCs is higher in the Bnc2-EGFP+ fraction, the absolute number of Bnc2+SSCs might not be higher than Bnc2-SSCs, since there are far more Bnc2- than Bnc2+ in the Lin-6C3-Thy1- gate. Better quantify both the percentage and absolute number of Bnc2+SSCs to clarify this.
2. In addition to bone marrow, osteoblast/osteocyte and periosteum, Prx1-Cre also deletes in chondrocytes and their descendants during endochondral ossification. Therefore, using Prx1-Cre to delete Bnc2 from the periosteum is not the best choice. In comparison, Ctsk-Cre is more suitable for this purpose.
3. Are there any bone phenotypes in Prx1-Cre; Bnc2-fl/fl mice without bone fractures? Characterization of their basal bone parameters is important for the reader to understand whether Bnc2 regulates skeletal development.

Minor points:

1. Line 38: "organs" should be changed to "organ";
2. Line 39: "reproduces" should be changed to "recapitulates";
3. Line 82: "specificity" should be changed to "specific";
4. Line 86: the second "SSCs" should be changed to "SSC", same for the rest of the manuscript when using it as an adjective (eg. line 92);
5. Line 94: mRNA-seq is not a standard terminology, better use "bulk RNA-seq";
6. Line 103: "Single cell sequencing" should be "Single-cell sequencing";
7. Line 241: "reduce" should be "increase"?
8. Line 298: "Which" should be in lowercase with a comma in front of it;
9. In Figure 2D, the left panel (snapshot) showed non-specific tdTomato signals in the bone marrow and skeletal muscle. Better replace by a new image;
10. In Figure 5H right column, Safranin O color is more brownish, which is different from the first two columns. Please show consistent staining images;
11. In Supplemental Figure 7A, incorrect color annotation for the EdU channel, which should be red, not cyan;
12. Method details are insufficient. For example, the details of "mRNA-seq" (Line 94) are found nowhere;
13. Please specify the full name of "BCSP" in Figures 1A, and "CB" in Figure 7E when they first appear;
14. The format for marking the age of mice is inconsistent. For example, in Figure 2C, treat tamoxifen at "8-week-old", in Figures 2E and 2G, treat tamoxifen at "3W, 4W". Please be consistent;
15. The micro-CT and X-Ray images lack scale bars.

Referee #2:

This manuscript demonstrates that Bnc2 is a marker of quiescent periosteal skeletal stem cells (SSCs) that clonally expand and contribute to fracture healing via endochondral ossification. Knockout of Bnc2 in Prx1-cre+ cells leads to impaired SSC proliferation and delayed bone healing. Mechanistically, BNC2 interacts with the NuRD complex, affecting histone H3 acetylation, and treatment with HDAC1/2 inhibitors partially rescues the defect. The identification of epigenetic modulation as a regulatory mechanism in skeletal repair and stem cell function is interesting. The study adds important insight to the field of skeletal stem cell biology by:

- Identifying Bnc2 as a functional periosteal SSC marker that is injury-inducible and contributes to endochondral, but not

intramembranous, ossification.

- Revealing an epigenetic mechanism (BNC2-NuRD interaction) that regulates SSC proliferation.
- Demonstrating that epigenetic drugs (HDAC inhibitors) can partially reverse repair defects in Bnc2-deficient mice, suggesting therapeutic potential.

However, several experimental inconsistencies and interpretative gaps should be addressed:

- The manuscript does not critically assess limitations, such as translational relevance of mouse models, potential off-target effects from the use of constitutive Prx1-cre (e.g., developmental defects or germline effects), or lack of identification of direct transcriptional targets of BNC2.
- The authors use fracture and bone drilling models with differing CT evaluation timepoints (7 days in Methods vs. 14 days in Supplemental Figure 3). The rationale for this inconsistency is not explained.
- The fracture end morphology at 3 dpf in Figure 1E is unclear and should be improved to convincingly show fracture status and periosteal expansion.
- The use of Prx1-cre instead of tamoxifen-inducible Prx1-ERT2-cre may result in unwanted developmental effects. Authors should justify this choice and discuss whether germline deletion or early knockout may influence the observed phenotypes.
- Safranin O/Fast Green staining shows inconsistent staining intensity in the third experimental group (right panel) of Figure 5H. This raises concerns about reproducibility and tissue processing.
- The tamoxifen injection schedule, route, and dosage are vaguely described. Moreover, injection timepoints are not clearly annotated in multiple figure panels (e.g., Figure 2A/C/E/G, 3A/F, 4A/C, 5A/B, Suppl. Fig. 5A).
- The study shows enhanced chondrogenic differentiation in vitro (Suppl. Fig. 8C-E) but impaired cartilage formation in vivo (Fig. 5H). This discrepancy is not discussed and may confuse readers about the functional role of BNC2.
- Rosa26-DTA mice are listed in the methods (line 311) but are not used in any experiment. This should be removed or clarified.
- Micro-CT is described as being conducted at 7 days post-op in Methods (line 330), but 14 days in Supplemental Fig. 3. This inconsistency should be resolved.
- Lastly, a summary figure integrating the main mechanistic findings (BNC2, NuRD, H3ac, SSC proliferation, and fracture healing outcomes) would greatly enhance clarity and accessibility for readers.

Referee #1:

In this study, Zhang et al. systematically explored the role of BNC2 in fracture healing. They identified BNC2⁺ cells as quiescent periosteal cells under homeostatic conditions, which undergo rapid expansion to promote endochondral ossification and fracture repair without affecting intramembranous ossification. Mechanistically, they found that BNC2 increases the acetylation levels of histone H3 through its interaction with the NuRD complex, thus promoting chromatin accessibility and transcriptional activation. Overall, this is an interesting study that uncovers a critical role of BNC2⁺ periosteal cells during fracture healing. To further support the authors' conclusions and strengthen the manuscript, there are a few concerns that need to be addressed in a potential revision:

Major points:

1. There is no sufficient ex vivo and in vivo data to support that BNC2 specifically marks periosteal stem cells. Therefore, better describe them as Bnc2⁺ periosteal cells throughout the manuscript, especially in the title, to avoid controversy. In Figures 1G and 1H, although the percentage of SSCs is higher in the Bnc2-EGFP⁺ fraction, the absolute number of Bnc2⁺SSCs might not be higher than Bnc2⁻SSCs, since there are far more Bnc2⁻ than Bnc2⁺ in the Lin⁻6C3⁻Thy1⁻ gate. Better quantify both the percentage and absolute number of Bnc2⁺SSCs to clarify this.

We thank the reviewer for this suggestion. Although our data indicated that Bnc2⁺ periosteal cells can fully participate in fracture repair (Figure 3), there is no sufficient ex vivo and in vivo data to support that BNC2 specifically marks periosteal stem cells. We describe the cells as Bnc2⁺ periosteal cells throughout the manuscript, especially in the title, to avoid controversy. Following the Reviewer's suggestions, we quantified both the percentage and absolute number of Bnc2⁺ SSCs. As shown in Figure 1H and 1I, the percentage of SSCs in Bnc2⁺ cells is much higher than that in Bnc2⁻ cells, however, the absolute number of Bnc2⁺ SSCs is less than the one of Bnc2⁻ SSCs since there are far more Bnc2⁻ than Bnc2⁺ in the Lin⁻6C3⁻Thy1⁻ gate.

Figure 1. (G) Flow cytometry clustering of SSCs in *BNC2-EGFP* negative and positive periosteal cells. **(H and I)** Quantitative statistics of percentage and absolute number of SSCs in *BNC2-EGFP* negative and positive periosteal cells. Data are presented as the means \pm SEM. n = 8. Unpaired t test.

2. In addition to bone marrow, osteoblast/osteocyte and periosteum, Prx1-Cre also deletes in chondrocytes and their descendants during endochondral ossification. Therefore, using Prx1-Cre to delete Bnc2 from the periosteum is not the best choice. In comparison, Ctsk-Cre is more suitable for this purpose.

We agree with the reviewer that it is a concern about the off-target of Prx1-Cre, which is expressed in the bone marrow, osteoblast/osteocyte, periosteum in homeostasis and chondrocytes and their descendants during endochondral ossification. Our data showed that BNC2 was hardly detected in the bone marrow and chondrocytes during homeostasis (Figure 2). To further address the concern of reviewer about the chondrocyte BNC2 in endochondral ossification, we isolated periosteal cells and performed chondrocyte differentiation. Loss of BNC2 did not affect the chondrocyte differentiation (Appendix Figure S9). And depletion of BNC2 with Prx1-Cre did not show cartilage phenotype in homeostasis (Appendix Figure S12). The reviewer also suggested the use of Ctsk-Cre for this purpose. Ctsk-Cre could label periosteal cells, however, Ctsk could also label osteoclasts. At the same time, Ctsk is also expressed in chondrocytes during the endochondral ossification of fracture repair (Debnath S., Yallowitz A. R. et al., 2018). The above analysis demonstrated that Ctsk-Cre mice might also have off-target issue. Due to time constraints related to revisions, we decided not to continue using Ctsk-Cre mice for the current study. However, to better understand the role of Bnc2 during bone fracture healing and to avoid the unwanted developmental effects, we did bone fracture healing analysis using Prx1-CreER; *Bnc2^{fl/fl}* mice (see below for Point 3).

Appendix Figure S9. (C) Alcian blue staining for chondrogenic differentiation of periosteal cell in *Prx1-cre; Bnc2^{fl/fl}* mice. (D and E) The expression of *Bnc2* and chondrogenic marker genes during chondrogenic differentiation of periosteal cell in *Prx1-cre; Bnc2^{fl/fl}* mice. Data are presented as the means \pm SEM. $n = 4$. Unpaired t test.

Appendix Figure S12. *Prx1-cre; Bnc2^{fl/fl}* mice did not show cartilage phenotype. (A) Micro-CT scan images of 6-month-old *Prx1-cre; Bnc2^{fl/fl}* mice. (B) SOFG staining of 6-month-old *Prx1-cre; Bnc2^{fl/fl}* mice.

3. Are there any bone phenotypes in *Prx1-Cre; Bnc2-fl/fl* mice without bone fractures?

Characterization of their basal bone parameters is important for the reader to understand whether *Bnc2* regulates skeletal development.

Following the reviewer's suggestions, we performed micro-CT scans on the femurs of 3-month-old *Prx1-Cre; Bnc2^{ff}* mice. The results showed that the trabecular bone mass of *Prx1-Cre; Bnc2^{ff}* mice decreased significantly (Appendix Figure S13A and B), but the cortical bone thickened obviously (Appendix Figure S13A and C). This indicates that *Bnc2* does regulate bone development in mice, manifested as positively regulating cancellous bone formation but negatively regulating cortical bone thickening. Besides, we used the *Prx1-CreER* to knock out *Bnc2* after adulthood (Appendix Figure S13D). Two months after the induction was completed, we performed micro-CT scans on the femurs of *Prx1-CreER; Bnc2^{ff}* mice. The results showed that there was no significant difference in bone mass among control and *Prx1-CreER; Bnc2^{ff}* mice (Appendix Figure S13E-G).

To better understand the role of *Bnc2* during bone fracture healing and to avoid the unwanted developmental effects, we did bone fracture healing analysis using *Prx1-CreER; Bnc2^{ff}* mice. Our results indicated that during the repair process in *Prx1-CreER; Bnc2^{ff}* mice, the callus volume decreased, but there was no significant difference in the callus volume at 35 dpf (Appendix Figure S4A-C). The micro-CT results showed that *Prx1-CreER; Bnc2^{ff}* mice had obvious nonunion of bone fracture (Appendix Figure 4D and E). This data indicated that the effects of *Bnc2* on bone fracture healing is not due to developmental defects.

Appendix Figure S13. Prx1-cre; Bnc2^{fl/fl} but not Prx1-creER; Bnc2^{fl/fl} mice showed decreased trabecular bone volume and thickened cortical bone. (A) Micro-CT scan images of cancellous bone and cortical bone of Prx1-cre; Bnc2^{fl/fl} mice. **(B and C)** Quantitative statistics of trabecular bone mass **(B)** and cortical bone **(C)** in Prx1-cre; Bnc2^{fl/fl} mice. Data are presented as the means ± SEM. n = 6. Unpaired t test. **(D)** Schematic diagram of tamoxifen induction strategy in Prx1-creER; Bnc2^{fl/fl} mice. **(E)** Micro-CT scan images of cancellous bone and cortical bone of Prx1-creER; Bnc2^{fl/fl} mice. **(F and G)** Quantitative statistics of trabecular bone mass **(F)** and cortical bone **(G)** in Prx1-creER; Bnc2^{fl/fl} mice. Data are presented as the means ± SEM. n = 4. Unpaired t test.

Appendix Figure S4. Prx1-creER; Bnc2^{fl/fl} mice showed significant impairment in fracture healing. (A) Schematic diagram of Prx1-creER; Bnc2^{fl/fl} mice fracture model. (B and C) X-Ray results (B) and quantitative statistics of callus index (C) in Prx1-creER; Bnc2^{fl/fl} mice at different stages after fracture. Data are presented as the means \pm SEM. n = 4. Unpaired t test. (D) micro-CT scanning results of Prx1-creER; Bnc2^{fl/fl} mice on 35 dpf. (E) Quantitative statistics of percentage of union and non-union fracture samples on 35 dpf in Prx1-creER; Bnc2^{fl/fl} mice. n = 4.

Minor points:

1. Line 38: "organs" should be changed to "organ";

We have changed the word "organ" in line 38 of the revised manuscript.

2. Line 39: "reproduces" should be changed to "recapitulates";

We have changed the word "recapitulates" in line 39 of the revised manuscript.

3. Line 82: "specificity" should be changed to "specific";

We have changed the word "specific" in line 80 of the revised manuscript.

4. Line 86: the second "SSCs" should be changed to "SSC", same for the rest of the manuscript when using it as an adjective (eg. line 92);

We have changed the word "SSC" in line 25, 85, 91, 208, 215 of the revised manuscript

5. Line 94: mRNA-seq is not a standard terminology, better use "bulk RNA-seq";

We have changed the word "bulk RNA-seq" in line 93 of the revised manuscript.

6. Line 103: "Single cell sequencing" should be "Single-cell sequencing";

We have changed the word "Single-cell sequencing" in line 101 of the revised manuscript.

7. Line 241: "reduce" should be "increase"?

We confirmed that it should be "reduce" in line 249 of the revised manuscript.

8. Line 298: "Which" should be in lowercase with a comma in front of it;

We have changed the word "which" in line 308 of the revised manuscript.

9. In Figure 2D, the left panel (snapshot) showed non-specific tdTomato signals in the bone marrow and skeletal muscle. Better replace by a new image;

We replaced the pictures in Figure 2D to reduce the non-specific signals.

10. In Figure 5H right column, Safranin O color is more brownish, which is different from the first two columns. Please show consistent staining images;

We re-performed the Safranin O and COL2 immunofluorescence staining and replaced Figures 5H and 5I.

Figure 5. (H) SOFG staining of Prx1-cre; Bnc2^{ff} and LepR-creER; Bnc2^{ff} mice on 7 dpf and 14 dpf. **(I)** COL2 immunofluorescence staining of Prx1-cre; Bnc2^{ff} and LepR-creER; Bnc2^{ff} mice on 7 dpf and 14 dpf.

11. In Supplemental Figure 7A, incorrect color annotation for the EdU channel, which should be red, not cyan;

We have changed the color annotation for the EdU channel in Appendix Figure S7A.

12. Method details are insufficient. For example, the details of "mRNA-seq" (Line 94) are found nowhere;

We have added the method descriptions of micromass culture, qRT-PCR, bulk RNA-seq and ATAC-seq in the "Methods" section. The details of sequencing sample collection can be found in the "Flow Cytometry" section of "Methods".

13. Please specify the full name of "BCSP" in Figures 1A, and "CB" in Figure 7E when they first appear;

We have specified the full name of "BCSP, bone cartilage stromal progenitor" in Figure 1A and "CB, cortical bone" in Figure 7E.

14. The format for marking the age of mice is inconsistent. For example, in Figure 2C, treat tamoxifen at "8-week-old", in Figures 2E and 2G, treat tamoxifen at "3W, 4W". Please be consistent;

We have changed "8-week-old" to "8W" in Figure 2C, "6-week-old" to "6W" in Figures 2E and 2G, "6-week-old" to "6W" in Figure 3A, "5-week-old" to "5W" in Figure 3F, "8-week-old" to "8W" in Figure 4A, "10-week-old" to "10W" in Figure 4C, "10-week-old" to "10W" in Figure 5B, "6-week-old" to "6W" in Figures 6A and 6D, "10-week-old" to "10W" in Appendix Figure S6A.

15. The micro-CT and X-Ray images lack scale bars.

We have added scale bars to the micro-CT and X-Ray images.

Referee #2:

This manuscript demonstrates that *Bnc2* is a marker of quiescent periosteal skeletal stem cells (SSCs) that clonally expand and contribute to fracture healing via endochondral ossification. Knockout of *Bnc2* in *Prx1-cre+* cells leads to impaired SSC proliferation and delayed bone healing. Mechanistically, BNC2 interacts with the NuRD complex, affecting histone H3 acetylation, and treatment with HDAC1/2 inhibitors partially rescues the defect. The identification of epigenetic modulation as a regulatory mechanism in skeletal repair and stem cell function is interesting. The study adds important insight to the field of skeletal stem cell biology by:

- Identifying *Bnc2* as a functional periosteal SSC marker that is injury-inducible and contributes to endochondral, but not intramembranous, ossification.
- Revealing an epigenetic mechanism (BNC2-NuRD interaction) that regulates SSC proliferation.
- Demonstrating that epigenetic drugs (HDAC inhibitors) can partially reverse repair defects in *Bnc2*-deficient mice, suggesting therapeutic potential.

However, several experimental inconsistencies and interpretative gaps should be addressed:

- The manuscript does not critically assess limitations, such as translational relevance of mouse models, potential off-target effects from the use of constitutive *Prx1-cre* (e.g., developmental defects or germline effects), or lack of identification of direct transcriptional targets of BNC2.

We thank the reviewer for the comments and agree with the reviewer that we did not critically assess limitations, we have supplemented this part of the content at the end of the "Discussion" section. The content is as follows:

“We found that *Bnc2*⁺ periosteal cells can fully participate in fracture repair, and the knockout of *Bnc2* in periosteal cells can lead to nonunion of fractures. However, since *Prx1-Cre* is continuously expressed during bone development, the use of the *Prx1-Cre; Bnc2*^{fl/fl} model cannot rule out the possible contribution of *Bnc2* in bone development to fracture nonunion. Therefore, we further demonstrated that *Bnc2* in periosteal cells of adult mice specifically regulates fracture repair without affecting bone homeostasis through *Prx1-CreER* mice. In addition, we found that BNC2 regulated the expression of genes related to proliferation and differentiation pathways

through interaction with the NuRD complex, but the key direct downstream target genes of BNC2 were not identified. And due to the wide range of action of HDAC inhibitors, while promoting fracture repair, HDAC inhibitors may have temporarily unknown side effects. Subsequently, we will further identify the target genes of BNC2 to achieve more precise regulation and provide new targets for the treatment of nonunion of bone fracture.”

-The authors use fracture and bone drilling models with differing CT evaluation timepoints (7 days in Methods vs. 14 days in Supplemental Figure 3). The rationale for this inconsistency is not explained.

We thank the reviewers for pointing out the mistake in the Methods. The drilling model was scanned by micro-CT 14 days after the operation, and the correction has been made in line 353. The bicortical bone fractures and bone drilling injury are two different models for bone regeneration. Bicortical bone fractures are primarily repaired by endochondral ossification with periosteal SSCs and the hard callus starts to form after 3-4 weeks post injury, so micro-CT scans are chosen to be performed 4-5 weeks after the operation. Bone drilling injuries were primarily repaired by intramembranous ossification with bone marrow SSCs and the new bone start to form after 2-3 days post injure, thus micro-CT scans are performed 2 weeks after the operation. In our study, BNC2 mainly expressed in the periosteal stem cells and play key roles in periosteal stem cell proliferation, and loss of BNC2 impaired the bicortical bone fractures.

-The fracture end morphology at 3 dpf in Figure 1E is unclear and should be improved to convincingly show fracture status and periosteal expansion.

We replaced the 3 dpf image in Figure 1E to show the morphology of periosteal expansion on the third day after the fracture more clearly.

Figure 1. (E and F) The expression of BNC2-EGFP in 0 dpf and 3 dpf and the quantitative statistics of the proportion of BNC2-EGFP positive cells. Data are presented as the means \pm SEM. n = 6. Unpaired t test.

-The use of Prx1-cre instead of tamoxifen-inducible Prx1-ERT2-cre may result in unwanted developmental effects. Authors should justify this choice and discuss whether germline deletion or early knockout may influence the observed phenotypes.

Following the reviewer's suggestions, to eliminate the influence of the extensive expression of Prx1-Cre during development, we used the Prx1-CreER mouse model to knock out Bnc2 after adulthood, thereby achieving the specific knockout of Bnc2 in periosteal cells in the adult stage. Our results indicated that during the repair process in Prx1-creER; Bnc2^{ff} mice, the callus volume decreased, but there was no significant difference in the callus volume at 35 dpf (Appendix Figure S4A-C). The micro-CT results showed that Prx1-creER; Bnc2^{ff} mice had an obvious nonunion of bone fracture (Appendix Figure S4D and E).

Appendix Figure S4. Prx1-creER; Bnc2^{ff} mice showed significant impairment in

fracture healing. (A) Schematic diagram of Prx1-creER; Bnc2^{ff} mice fracture model. (B and C) X-Ray results (B) and quantitative statistics of callus index (C) in Prx1-creER; Bnc2^{ff} mice at different stages after fracture. Data are presented as the means ± SEM. n = 4. Unpaired t test. (D) micro-CT scanning results of Prx1-creER; Bnc2^{ff} mice on 35 dpf. (E) Quantitative statistics of percentage of union and non-union fracture samples on 35 dpf in Prx1-creER; Bnc2^{ff} mice. n = 4.

-Safranin O/Fast Green staining shows inconsistent staining intensity in the third experimental group (right panel) of Figure 5H. This raises concerns about reproducibility and tissue processing.

We re-performed the Safranin O and COL2 immunofluorescence staining and replaced Figures 5H and 5I.

Figure 5. (H) SOFG staining of Prx1-cre; Bnc2^{ff} and LepR-creER; Bnc2^{ff} mice on 7 dpf and 14 dpf. (I) COL2 immunofluorescence staining of Prx1-cre; Bnc2^{ff} and LepR-creER; Bnc2^{ff} mice on 7 dpf and 14 dpf.

-The tamoxifen injection schedule, route, and dosage are vaguely described. Moreover, injection timepoints are not clearly annotated in multiple figure panels (e.g., Figure 2A/C/E/G, 3A/F, 4A/C, 5A/B, Suppl. Fig. 5A).

We have supplemented the injection information of tamoxifen in the Figures, including the injection method, dose and schedule.

-The study shows enhanced chondrogenic differentiation in vitro (Suppl. Fig. 8C-E) but impaired cartilage formation in vivo (Fig. 5H). This discrepancy is not discussed and may confuse readers about the functional role of BNC2.

We thank the reviewer for the comments. In Appendix Figure S8C-E (Appendix Figure S9C-E in revised version), only a mild increase in the expression of COL2 was detected, alcian blue staining did not significantly enhance, and there was no difference in the expression of SOX9, a key transcription factor for chondrocyte differentiation. We also added the detection of Aggrecan expression, which showed no significant difference. And depletion of BNC2 with Prx1-Cre did not show cartilage phenotype in homeostasis (Appendix Figure S12). In addition, we found that the absence of Bnc2 led to a significant decrease in the proliferation ability of periosteal cells (Figure 6). Therefore, we believe that the cartilage formation disorder during the fracture repair process in Prx1-Cre; Bnc2^{fl/fl} mice is mainly caused by the obstruction of precursor cell proliferation.

Appendix Figure S9. (C) Alcian blue staining for chondrogenic differentiation of periosteal cell in Prx1-cre; Bnc2^{fl/fl} mice. (D and E) The expression of Bnc2 and chondrogenic marker genes during chondrogenic differentiation of periosteal cell in Prx1-cre; Bnc2^{fl/fl} mice. Data are presented as the means ± SEM. n = 4. Unpaired t test.

Appendix Figure S12. Prx1-cre; Bnc2^{ff} mice did not show cartilage phenotype. (A) Micro-CT scan images of 6-month-old Prx1-cre; Bnc2^{ff} mice. (B) SOFG staining of 6-month-old Prx1-cre; Bnc2^{ff} mice.

-Rosa26-DTA mice are listed in the methods (line 311) but are not used in any experiment. This should be removed or clarified.

We have removed the information of the Rosa26-DTA mice.

-Micro-CT is described as being conducted at 7 days post-op in Methods (line 330), but 14 days in Supplemental Fig. 3. This inconsistency should be resolved.

We thank the reviewers for pointing out that we made a mistake in the "Method". The drilling model was scanned by micro-CT 14 days after the operation, and the correction has been made in line 350 of the revised manuscript.

-Lastly, a summary figure integrating the main mechanistic findings (BNC2, NuRD, H3ac, SSC proliferation, and fracture healing outcomes) would greatly enhance clarity and accessibility for readers.

We thank the reviewer for this suggestion. We made a pattern diagram indicating that BNC2 regulates the proliferation of periosteal cells through interaction with the NuRD complex, thereby promoting fracture repair in mice, as shown in Appendix Figure S11.

Appendix Figure S11. Working model of BNC2 promoting fracture repair. (A) BNC2 regulates the proliferation of periosteal cells through interaction with the NuRD complex, thereby promoting fracture repair in mice.

Reference

Debnath S., Yallowitz A. R., McCormick J., Lalani S., Zhang T., Xu R., Li N., Liu Y., Yang Y. S., Eiseman M., Shim J. H., Hameed M., Healey J. H., Bostrom M. P., Landau D. A., B. GM (2018) Discovery of a periosteal stem cell mediating intramembranous bone formation. *Nature* 562: 133-139

Dear Dr Zou, dear Dr Wang,

Thank you for submitting your revised manuscript (EMBOJ-2025-121039R) to The EMBO Journal, as well for your patience with our feedback. Your amended study was sent back to the referees for their scientific reassessment, and we have received reports from both of them, which I enclose below. As you will see, the reviewers state that the work has been substantially enhanced by the revisions and they are now broadly in favour of publication.

Thus, we are pleased to inform you that your manuscript has been accepted in principle for publication in The EMBO Journal.

We now need you to take care of a number of issues related to formatting and data presentation as detailed below, which should be addressed at re-submission.

As you might have seen on our web page, every paper at the EMBO Journal now includes a 'Synopsis', displayed on the html and freely accessible to all readers. The synopsis includes a 'model' figure as well as 2-5 one-short-sentence bullet points that summarize the article. I would appreciate if you could provide this figure and the bullet points.

Thank you again for giving us the chance to consider your manuscript for The EMBO Journal, I look forward to hearing from you and receiving your final revised version of the manuscript.

Best regards,

Daniel Klimmeck

>> Authors: All corresponding authors need accounts linked to an ORCID number entered into our online system (W.Z., L.W.). Please see below for additional information.

>> Author Contributions: Remove the author contributions information from the manuscript text. Note that CRediT has replaced the traditional author contributions section as of now because it offers a systematic machine-readable author contributions format that allows for more effective research assessment. and use the free text boxes beneath each contributing author's name to add specific details on the author's contribution.

More information is available in our guide to authors.
<https://www.embopress.org/page/journal/14602075/authorguide>

>> Enter a 'Disclosure and Competing Interests Statement' into the manuscript.

>> Correct the order of the manuscript sections as follows: Abstract / Keywords / Introduction / Results / Discussion / Methods / Data Availability / Acknowledgements / Disclosure and Competing Interests Statement / References / Main Figure Legends / Tables / Expanded View Figure Legends. The Data Availability section should be after the Methods.

>> Remove the explicit 'Limitations of the study' section and integrate into the Discussion.

>> Funding: please enter the following funding information into our online system: ' National Natural Science Foundation of

China (82530082, 82230082, 82102554), Space Medical Experiment Project of China Manned Space Program (HYZHXM01025), and the Strategic Priority Research Program of the Chinese Academy of Science (grant XDB0570000) '.

>> References: adjust the reference format to EMBO Journal format, 10 authors et al. .

>>Appendix file with ToC: compile all appendix figures in one PDF and add a table of contents with page numbers, please remove the appendix figure legends from the main manuscript text..

>> Figure callouts: Please ensure that the figures and panels are called out in sequential order. Currently, Appendix Figure S12 is called out before Appendix Figure S10.

>> Add a Reagents and Tools table to the Methods section, as a separate file using the existing template in the Guide For Authors, listing key reagents, experimental models, software and relevant equipment.

>> Data availability section: please provide database annotation for the RNA-seq dataset. Add hyperlinks to the datasets and make sure they are publicly accessible.

>> Consider additional changes and comments from our production team as indicated below:

- Figure legends:

1. Please note that the exact p values are not provided in the legends of figures 1F, H, I; 6H, I; 7G, I.
2. Please indicate the statistical test used for data analysis in the legend of figure 7L
3. Please note that n=2 in figure 1C4.
4. Please note that the dotted borders are not defined in the legend of figures 1E, 4F. This needs to be rectified.

Please note that as of January 2016, our new EMBO Press policy asks for corresponding authors to link to their ORCID iDs. You can read about the change under "Authorship Guidelines" in the Guide to Authors here: <http://emboj.embopress.org/authorguide>

In order to link your ORCID iD to your account in our manuscript tracking system, please do the following:

1. Click the 'Modify Profile' link at the bottom of your homepage in our system.
2. On the next page you will see a box half-way down the page titled ORCID*. Below this box is red text reading 'To Register/Link to ORCID, click here'. Please follow that link: you will be taken to ORCID where you can log in to your account (or create an account if you don't have one)
3. You will then be asked to authorise Wiley to access your ORCID information. Once you have approved the linking, you will be brought back to our manuscript system.

We regret that we cannot do this linking on your behalf for security reasons. We also cannot add your ORCID iD number manually to our system because there is no way for us to authenticate this iD number with ORCID.

Thank you very much in advance.

Referee #1:

The authors have successfully addressed my concerns. The manuscript is now ready to be accepted for publication.

Referee #2:

I believe the authors have adequately addressed all concerns raised during the review process. They have made substantial improvements to the manuscript, including:

- Clarifying the terminology regarding BNC2⁺ periosteal cells.
- Providing new data using the inducible Prx1-CreER model to establish the specific role of BNC2 in fracture healing, independent of developmental effects.
- Adding quantitative analyses and appropriate controls as requested.
- Improving the quality and consistency of the figures.
- Expanding the discussion of the study's limitations and its translational potential.

The study identifies a novel and important role for BNC2 in activating periosteal stem cells for bone repair through an epigenetic mechanism. The mechanistic link to the NuRD complex and the partial rescue with HDAC inhibitors are particularly compelling, highlighting potential therapeutic avenues.

Overall, the evidence presented is now robust and convincingly supports the conclusions.

The authors addressed the remaining editorial issues.

Dear Dr Zou, dear Dr Wang,

Thank you for submitting the revised version of your manuscript. I have now evaluated your amended manuscript and concluded that the remaining minor concerns have been sufficiently addressed.

I am thus pleased to inform you that your manuscript has been accepted for publication in the EMBO Journal.

Best regards,

Daniel Klimmeck

Daniel Klimmeck, PhD
Senior Editor
The EMBO Journal
EMBO
Postfach 1022-40
Meyerhofstrasse 1
D-69117 Heidelberg
contact@embojournal.org

Please note that it is The EMBO Journal policy for the transcript of the editorial process (containing referee reports and your response letters) to be published as an online supplement to each paper. If you should prefer removal of any referee-only figures included in the point-by-point response(s), e.g. because they may still be used for future publication or because they have been reproduced from published work by others, please do let us know immediately via response email.

More information is available here: https://www.embopress.org/transparent-process#Review_Process